# Crystal structures and insights into precursor tRNA 5'-end processing by prokaryotic minimal protein-only RNase P

Yangyang Li[1], Shichen Su[2], Yanqing Gao [1], Guoliang Lu[2], Hehua Liu[1], Xi Chen[1], Zhiwei Shao[1], Yixi Zhang[1], Qiyuan Shao[1], Xin Zhao[1], Jie Yang[1], Chulei Cao[1], Jinzhong Lin [2], Jinbiao Ma [2] & Jianhua Gan [1✉]

Besides the canonical RNA-based RNase P, pre-tRNA 5'-end processing can also be catalyzed by protein-only RNase P (PRORP). To date, various PRORPs have been discovered, but the basis underlying substrate binding and cleavage by HARPs (homolog of Aquifex RNase P) remains elusive. Here, we report structural and biochemical studies of HARPs. Comparison of the apo- and pre-tRNA-complexed structures showed that HARP is able to undergo large conformational changes that facilitate pre-tRNA binding and catalytic site formation. *Planctomycetes bacterium* HARP exists as dimer in vitro, but gel filtration and electron microscopy analysis confirmed that HARPs from *Thermococcus celer*, *Thermocrinis minervae* and *Thermocrinis ruber* can assemble into larger oligomers. Structural analysis, mutagenesis and in vitro biochemical studies all supported one cooperative pre-tRNA processing mode, in which one HARP dimer binds pre-tRNA at the elbow region whereas 5'-end removal is catalyzed by the partner dimer. Our studies significantly advance our understanding on pre-tRNA processing by PRORPs.

[1] Shanghai Public Health Clinical Center, State Key Laboratory of Genetic Engineering, Collaborative Innovation Center of Genetics and Development, Department of Biochemistry and Biophysics, School of Life Sciences, Fudan University, Shanghai 200438, China. [2] State Key Laboratory of Genetic Engineering, Collaborative Innovation Center of Genetics and Development, Department of Biochemistry, School of Life Sciences, Fudan University, Shanghai 200438, China. ✉email: ganjhh@fudan.edu.cn

Transfer RNA (tRNA) decodes mRNA and transfers the genetic information from mRNA to protein. The precursor of tRNA (pre-tRNA) undergoes a series of post-transcriptional processes, including nucleobase modification[1–4], splicing[5–7], 5'-end cleavage[8–10] and 3'-end cleavage and conserved CCA addition[11–14]. Correct processing is essential for the function of tRNA. In addition to pre-tRNA coding genes, mutations in many pre-tRNA processing enzymes have also been linked with serious diseases[15–21]. RNase P catalyzes the 5'-end cleavage of pre-tRNA. The canonical RNase P was first discovered in Bacteria[22,23], it was later found in Archaea and Eukarya[24]. The canonical RNase P occurs as ribonucleoprotein (RNP) composed of one RNA and various number of proteins[24,25], but pre-tRNA 5'-end is all cleaved by the RNA molecule.

Interestingly, recent studies showed that pre-tRNA 5'-end can also be cleaved by protein-only RNase Ps (PRORP). To date, PRORP has been found in the mitochondria of human, *Caenorhabditis elegans* (*C. elegans*), *Drosophila* and many other animals[26,27]. These mitochondrial PRORPs are composed of three proteins, MRPP1-3. Pre-tRNA 5'-end cleavage is catalyzed by MRPP3 (Fig. 1A), but it strictly requires the assistance of MRPP1 and MRPP2. PRORP is also present in plants and protists[28], such as *Arabidopsis thaliana* (*A. thaliana*) and *Trypanosoma brucei* (*T. brucei*)[29]. Although *At*PRORPs and *Tb*PRORPs are homologous to human MRPP3 (Fig. 1A), they are active on their own.

Bacteria and Archaea do not express eukaryotic-like PRORP proteins, but in vitro pre-tRNA 5'-end cleavage activity has been confirmed for protein Aq_880 (Fig. 1A) encoded by the hyperthermophilic bacterium *Aquifex aeolicus* (*A. aeolicus*)[30]. Besides *A. aeolicus*, bioinformatic analysis showed that homologs of Aquifex RNase P (termed HARP hereafter) are also present in some other Bacteria and are even more wide-spread in Archaea[30,31]. Like the eukaryotic PRORPs, HARPs are members of the PIN domain superfamily, but belong to a different subgroup (PIN_5 cluster)[32]. Eukaryotic PRORP contains three domains: the pentatricopeptide repeat (PPR) domain, the central domain and the catalytic metallonuclease domain. Compared to eukaryotic PRORP, HARP is much shorter and it lacks both PPR and the central domains. The catalytic metallonuclease domain is conserved, but it only shares very limited sequence similarity (<25%) between HARP and eukaryotic PRORP.

In addition to Aq_880, the in vitro pre-tRNA 5'-end cleavage activity has also been confirmed for HARPs from *Haloferax volcanii* (*H. volcanii*) and *Matheanosarcina mazei* (*M. mazei*), which are members of the euryarchaeotes[33]. Aq_880 shares conserved residues with *Hv*HARP, *Mm*HARP and HARPs from many other species (Supplementary Fig. 1A). Knockout of the HARP gene showed no growth defects in *H. volcanii* and *M. mazei*, indicating that the RNA-based RNase P is the main and essential RNase P in these archaea[33]. Different from euryarchaeotes, HARP is the only RNase P expressed in *A. aeolicus* and many related *Aquificaceae*, such as *Hydrogenobacter thermophiles* and *Thermocrinis albus* DSM. Aq_880 is able to replace the RNase P complex in budding yeast *Saccharomyces cerevisiae* and rescue the growth of *E. coli* strain BW, in which the chromosomal expression of RNase P RNA is switched off by replacing arabinose with glucose in the medium[30]. To improve our understanding on pre-tRNA processing by HARP, we performed extensive structural and biochemical studies. Here, we report the crystal structures of *Planctomycetes bacterium* HARP (*Pb*HARP) and *Thermococcus celer* HARP (*Tc*HARP). Comparison of the apo- and pre-tRNA-complexed *Pb*HARP structures showed that HARP is able to undergo large conformational changes that facilitate pre-tRNA binding and catalytic site formation. Electron microscopy analysis of *Thermocrinis ruber* HARP (*Tr*HARP) showed that the dimer of HARP can assemble into oligomers. Structural analysis, mutagenesis, in vitro cleavage and cross-linking assays all suggested a HARP pre-tRNA binding and cleavage mode involving the cooperation of two dimers.

## Results

**Pre-tRNA 5'-end cleavage activity is conserved in HARPs.** To investigate whether the pre-tRNA 5'-end cleavage activity is conserved in HARPs, we constructed and purified four HARPs, *Pb*HARP, *Tc*HARP, *Tm*HARP and *Tr*HARP, which are coded by *Planctomycetes bacterium* GWF2_40_8 (*P. bacterium*), *Thermococcus celer* (*T. celer*), *Thermocrinis minervae* (*T. minervae*) and *Thermocrinis ruber* (*T. ruber*), respectively. *T. celer* is a hyperthermophilic archaeon, *T. minervae* and *T. ruber* are all members of *Aquificaceae*. *Pb*HARP, *Tc*HARP, *Tm*HARP and *Tr*HARP share 60-80% sequence similarities with Aq_880, and the sequence identity is about 30% among the five proteins (Supplementary Fig. 1B).

All proteins were purified to homogeneity. During purification, we noticed that the oligomerization state of HARP varies (Supplementary Fig. 2). *Pb*HARP exists as dimer in buffer containing 300–500 mM salt; at lower concentration of salt, it precipitates out. The oligomerization state of *Tc*HARP is changeable. At higher concentration of salt (200–500 mM), it exists as dimer; however, it assembles into larger oligomers in buffer containing 100 mM salt. Using 1.0 μM *Thermus thermophiles* pre-tRNA$^{Gly}$ (Supplementary Fig. 3A) purified in the laboratory, we performed in vitro cleavage assay (Fig. 1B) for *Pb*HARP, *Tc*HARP, *Tm*HARP, *Tr*HARP and Aq_880, which served as a positive control. The HARP concentrations are all fixed at 0.05 μM. Like Aq_880, other four HARPs can also cleave the 5'-end off from the pre-tRNA substrate. At a reaction time of 20 min, about 74%, 79% and 48% pre-tRNA$^{Gly}$ was cleaved by *Tc*HARP, *Tm*HARP and *Tr*HARP, respectively. Although the cleavage activity of *Pb*HARP is relatively weak, it can cleaves about 6% pre-tRNA$^{Gly}$ at a reaction time of 20 min.

**Overall folding and dimerization of apo- *Pb*HARP and *Tc*HARP.** Puzzled by their different oligomerization states and catalytic efficiencies, we performed crystallographic studies for all the HARPs. No crystal grew for Aq_880, *Tm*HARP or *Tr*HARP, but we successfully solved the apo-form structures of *Pb*HARP and *Tc*HARP at atomic resolution (Supplementary Table 1). *Pb*HARP crystal belongs to P3$_1$ space group, per asymmetric unit contains two *Pb*HARP dimers. *Tc*HARP crystal belongs to P2$_1$2$_1$2$_1$ space group and there is only one *Tc*HARP dimer in the asymmetric unit. Both *Pb*HARP and *Tc*HARP are of α/β fold in nature (Fig. 1C-F, Supplementary Fig. 4); they share six β-strands (β1-β6) and ten α-helices (α1-α10). Compared to *Tc*HARP, *Pb*HARP is 14 amino acid longer at its N-terminus, which forms one short extra helix, α0. β1-β3 and β5-β6 are parallel to each other, forming one flat β-sheet at the center of the metallonuclease domain. Helices α1-α6 and α10 reside on one side of the β-sheet, whereas α9 and the C-terminus of α8 locate on the opposite site.

β4 and α7 are not involved in the metallonuclease domain formation. β4 is linked to the metallonuclease domain through one short linker that is perpendicular to the C-terminus of α8 (Fig. 1C). α7 is separated from α8 by a short linker. Indicated by the weak electron density and different conformations (Supplementary Fig. 4C), the α7-α8 connecting linker is flexible. Both α7 (amino acids 97-122) and α8 (amino acids 131-165) are very long, forming a four-helix bundle (HB) with α7 and α8 from the partner molecule. Due to the presence of Lys or Arg residues, the surface of HB is highly positive in charge (Fig. 1D). The

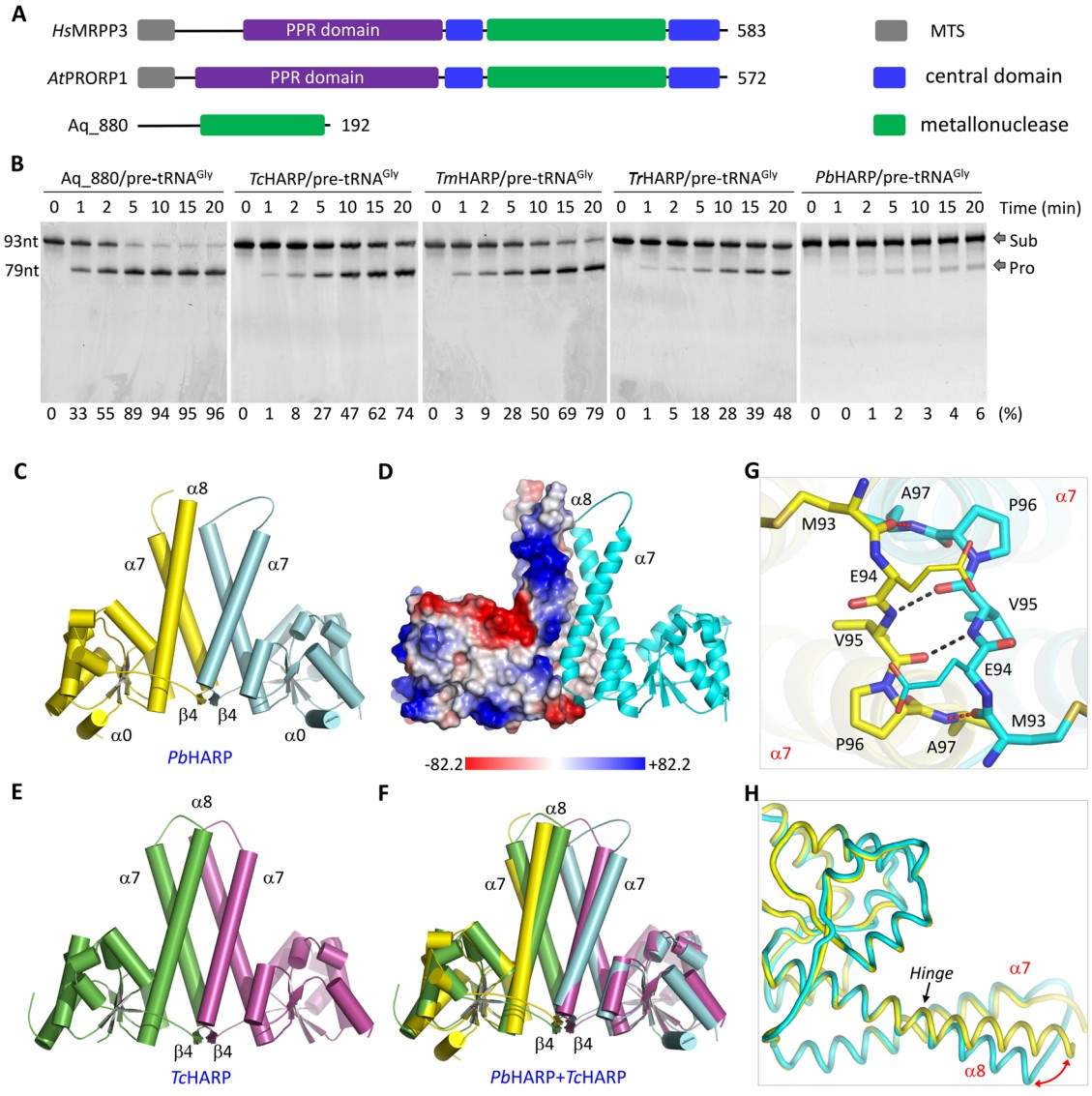

**Fig. 1 pre-tRNA 5'-end processing and folding of HARPs. A** Domain architectures of PRORPs. **B** In vitro pre-tRNA^Gly cleavage assays catalyzed by different HARPs. The substrate cleavage percentage (%) is shown at the bottom of the gels. Experiments were repeated independently twice with similar results. Source data are provided in Source Data file. **C, D** Cartoon and surface presentation showing the folding of PbHARP AB dimer observed in the apo-form structure. **E** Cartoon presentation of the apo-form TcHARP structure. **F** Structural superposition of PbHARP and TcHARP. The PbHARP AB dimer is colored as in panel (**C**), the TcHARP dimer is colored as in panel (**E**). **G** H-bond interactions involved in PbHARP dimerization in the apo-form structure. **H** Structural superposition showing the conformational difference between the HB domains of PbHARP A and B monomers. In panels (**G**), (**H**), the C-atoms are colored in yellow and cyan in the A and B monomers, respectively.

metallonuclease domain is also positive on the surface distal from HB, whereas is highly negative at the region facing HB.

β4 plays an important role in PbHARP dimerization. As depicted in Fig. 1G, the β4 strands of the two monomers form four hydrogen (H) binding interactions: two between the backbone carbonyl and amino functions of the two V95 residues and the other two mediated by M93 and A97 residues. These H-bond interactions are very stable, supported by their short distances (2.7–2.85 Å). The dimerization manner of TcHARP is similar to that of PbHARP. However, structural analysis revealed some conformational differences between the PbHARP and TcHARP dimers, especially in the HB regions (Fig. 1F). Further analysis showed that the overall folds of the two TcHARP monomers are similar to that of PbHARP monomer A (or C) (Supplementary Fig. 4D), the conformational differences are mainly caused by the bending of α8 of PbHARP monomer B (or D), hinged at residue E147 (Fig. 1H). The root-mean-square deviation (RMSD)

value between TcHARP and PbHARP A or C monomers is 1.0 Å, whereas is 1.4 Å between TcHARP and PbHARP B or D monomers.

**Pre-tRNA recognition by PbHARP.** Besides apo- proteins, we also performed co-crystallization trials using pre-tRNA^Gly, but no crystal was obtained. We then transcribed and purified another three pre-tRNAs, including E. coli pre-tRNA^His, mitochondrial pre-tRNA^Cys and chloroplastic pre-tRNA^Phe (Supplementary Fig. 3A). After extensive trials, we solved the complex structure of PbHARP/pre-tRNA^His at 2.8-Å resolution (Supplementary Table 1). The crystal belongs to $P4_12_12$ space group and it contains two PbHARP dimers (AB and CD) and one pre-tRNA^His molecule in the asymmetric unit (Fig. 2A). As indicated by the low RMSD value (0.6 Å), the overall folds of PbHARP AB and CD dimers are very similar (Supplementary Fig. 5A). Out of the 85 nucleotides of pre-tRNA^His, G-10 to U-2, U32 to A36 and C73/

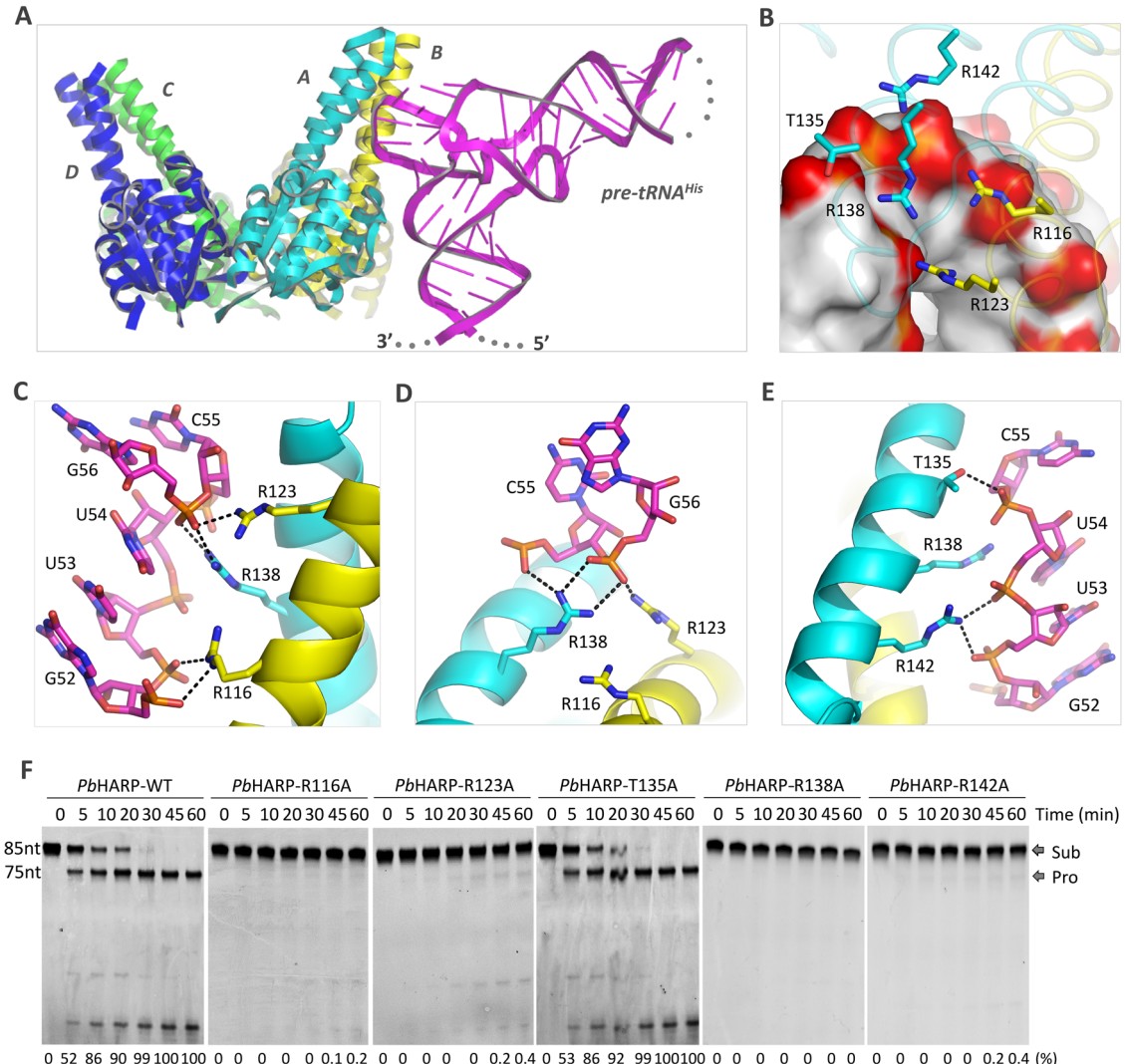

**Fig. 2 The complex structure of *Pb*HARP/pre-tRNA^His.** **A** Cartoon presentation showing the overall fold and assembly of *Pb*HARP/pre-tRNA^His complex. **B** The relative orientation between pre-tRNA^His and the conserved Arg residues. Pre-tRNA^His is shown as surface, the phosphate backbones are colored in red. **C**–**E** The detailed interactions between pre-tRNA^His TψC-loop and *Pb*HARP AB dimer. Pre-tRNA^His and all the interacting Arg and Thr residues are shown as sticks. The C-atoms of pre-tRNA^His, the A and B monomers of *Pb*HARP are colored in magenta, yellow, and cyan, respectively. **F** In vitro pre-tRNA^His cleavage assays catalyzed by wild-type and mutated *Pb*HARP. The protein and substrate concentrations are all fixed at 1 μM. The substrate cleavage percentage (%) is shown at the bottom of the gels. Experiments were repeated independently three times with similar results. Source data are provided in Source Data file.

A74 located at the 5'-end, anticodon loop and the 3'-end are disordered, whereas other nucleotides are all well-defined (Supplementary Fig. 5B).

Pre-tRNA^His is recognized by the AB dimer of *Pb*HARP. Although the overall folds of the metallonuclease domains and the HB domains are conserved, their relative orientations are different within the two monomers (Supplementary Fig. 5C). The asymmetric dimerization orients the side chains of four HB domain Arg residues, including R116 and R123 of monomer B and R138 and R142 of monomer A, toward the pre-tRNA^His TψC-loop where they form extensive H-bond interactions with the phosphate backbones of G52-G56 region (Fig. 2B). In details, R116 forms two H-bonds: one with the OP1 atom of G52 and the other with the OP2 atom of U53 (Fig. 2C). Both R123 and R138 interact with G56. Unlike G52-C55 region, the phosphate backbone of G56 is severely bent and packs against the nucleobase of U54 (Fig. 2C). R123 interacts with the OP1 atom of G56, whereas R138 interacts with both OP1 and OP2 atoms. Besides G56, R138 also interacts with the OP2 atom of C55

(Fig. 2D). R142 forms two H-bonds: one with the OP1 atom of U53 and the other with the OP2 atom of U54 (Fig. 2E).

In the complex structure, the acceptor stem and the predicted 5'-end cleavage site of pre-tRNA^His are far away from *Pb*HARP (Fig. 2A), which prompted us to ask whether *Pb*HARP has pre-tRNA^His cleavage activity. To this end, we carried out in vitro cleavage assay using 0.05 μM *Pb*HARP and 1.0 μM pre-tRNA^His (Supplementary Fig. 3B). Although it is weak, *Pb*HARP can remove the 5'-end of pre-tRNA^His. Besides pre-tRNA^His, we showed that *Pb*HARP is also able to process pre-tRNA^Cys and pre-tRNA^Phe under identical reaction conditions (Supplementary Fig. 3B). We then asked whether the pre-tRNA cleavage by *Pb*HARP is specific. To address this question, we performed pre-tRNA^Phe (5.0 μM) cleavage assay using 0.25 μM *Pb*HARP, Aq_880 or *E. coli* RNase P. The reaction mixtures were separated on 20% denaturing UREA-PAGE gels (Supplementary Fig. 3C). No cleavage product was observed in the absence of RNase P. Two 5'-leader cleavage products were observed, one being identical in size to that generated by Aq_880 and *E. coli* RNase

P (Supplementary Fig. 3C, outer right lane). This shows that *Pb*HARP cleaves pre-tRNA at the canonical cleavage site in addition to some apparent miscleavage at the neighboring upstream site.

To verify whether the interactions observed in the complex are functionally relevant, we constructed five *Pb*HARP mutants, including R116 A, R123A, T135A, R138A and R142A. T135 interacts with pre-tRNA$^{His}$ in the complex (Fig. 2E), but it is not conserved (Supplementary Fig. 1). Surprisingly, in vitro binding assays showed that Ala substitution of the five pre-tRNA recognizing residue has no obvious impacts on pre-tRNA binding by *Pb*HARP (Supplementary Fig. 6A). Unlike many other homology proteins, *Pb*HARP is longer at its N-terminus. The extra *Pb*HARP terminus is highly positive in charge; out of the total 13 residues, 7 are Arg or Lys (Supplementary Fig. 1B). To test whether the extra terminus contributes to the unexpected pre-tRNA binding results, we constructed one truncated protein, *Pb*HARP_ΔN (aa 14-203). Compared to the full-length protein, the pre-tRNA binding ability of *Pb*HARP_ΔN is weaker (Supplementary Fig. 6B). Started from *Pb*HARP_ΔN, we then constructed several *Pb*HARP mutants with pre-tRNA recognizing residue mutation. Although it has no strong impact on the overall pre-tRNA binding ability, Ala substitution of either R116 (for the *Pb*HARP_ΔN _R116A mutant) or R123 (for the *Pb*HARP_ΔN _R123A mutant) altered the pre-tRNA binding mode of *Pb*HARP, indicated by the smeared bands on the gel. The pre-tRNA binding ability of *Pb*HARP_ΔN _R138A is weaker than that of *Pb*HARP_ΔN. No obvious difference was observed for the *Pb*HARP_ΔN_R142A mutant, but the *Pb*HARP_ΔN_R138A/R142A mutant showed weaker pre-tRNA binding ability, compared to the *Pb*HARP_ΔN _R138A mutant.

The above mutation and binding assay results indicated that R116, R123, R138 and R142 affect pre-tRNA binding by *Pb*HARP. To further confirm the functional importance of these residues, we performed in vitro cleavage assay using equimolar (1.0 μM) pre-tRNA$^{His}$ and proteins (Fig. 2F). Substitution of T135 by Ala has no obvious impacts on pre-tRNA$^{His}$ cleavage, compared to the wild-type (WT) *Pb*HARP. In contrast to T135, R116, R123, R138 and R142 are all highly conserved (Supplementary Fig. 1). Ala substitution of either R116, R123 or R142 caused a dramatic reduction of the protein's pre-tRNA$^{His}$ cleavage activity. No detectable pre-tRNA$^{His}$ cleavage activity at all was observed for the R138A mutant. Altogether, these observations suggested that R116, R123, R138 and R142 are critical for assembly of the catalytically active *Pb*HARP/pre-tRNA complex.

**Domain rearrangement and active site assembly of *Pb*HARP.** HARP contains one metallonuclease domain, whose activity is normally dependent on cations[34]. Previous sequence analysis and mutagenesis studies[30] suggested that the catalytic center of Aq_880 is composed of three aspartate residues, corresponding to D151, D155 and D173 in *Pb*HARP (Supplementary Fig. 1B). The functional importance of these catalytic residues could be confirmed by the in vitro cleavage assays using equimolar (1 μM) protein and pre-tRNA$^{His}$ (Supplementary Fig. 7). *Pb*HARP has no pre-tRNA$^{His}$ cleavage activity in the absence of divalent cation. In the presence of Mg$^{2+}$ (the common cofactor of metallonucleases), *Pb*HARP can efficiently remove pre-tRNA$^{His}$ 5'-end. At a reaction time of 5 min, about 60% of the substrate were cleaved; only trace amounts of intact substrate remained at a reaction time of 20 min. In contrast to WT *Pb*HARP, no pre-tRNA$^{His}$ cleavage activity was observed for the D151A, D155A and D173 A mutants (Supplementary Fig. 7).

In the apo-form *Pb*HARP structure, both D151 and D155 reside at the central region of α8 and D173 locates at the N-terminus of α9 (Fig. 3A, Supplementary Fig. 8A). The side chains of D155 and D173 are close to each other, forming water-mediated H-bond interactions. In contrast, the side chain of D151 points away from D155, the averaged distance between their carboxylate groups exceeding 7 Å. R110 locates at the middle of α7; its side chain points toward D155, forming two direct H-bond interactions between their guanidine and carboxylate groups. Such conformations are incompatible with Mg$^{2+}$ coordination with D151, D155 and D173, suggesting that the apo-form *Pb*HARP structure is actually in an inactive state.

As demonstrated for many nucleases, such as RNase III[35,36], RNase D[37] and RNase H[38], Ca$^{2+}$ is insufficient in supporting the cleavage activity but can mimic Mg$^{2+}$ in coordination. The crystal of *Pb*HARP/pre-tRNA$^{His}$ was grown in the presence of 10 mM CaCl$_2$. In the structure, several well-defined Ca$^{2+}$ ions were captured (Supplementary Fig. 8B). As demonstrated by the A monomer of *Pb*HARP (Fig. 3B), Ca$^{2+}$ directly coordinates with the side chain OD1 atom of D151. In addition, it also coordinates with the main chain O atoms of R146 and I149. The side chains of D151, D155 and D173 are close to each other. Although not observed in the structure, this arrangement may allow *Pb*HARP to coordinate a second divalent cation, thereby utilizing the common two metal ion-assisted mechanism of catalysis[35,37,38].

Divalent metal ion coordination, substrate binding and mutagenesis studies all indicated that *Pb*HARP in the complex is in the active state. To reveal the structural basis underlying the inactive and active state switching, we compared the apo- and the pre-tRNA$^{His}$-complexed *Pb*HARP structures. As depicted in Fig. 3C, the relative orientations of the metallonuclease domains are preserved, but the orientations of the HB domains are significantly different in the two structures. Compared to the apo-structure, all the HB domain helices are anticlockwisely rotated in the complex. Further structural analysis (Supplementary Fig. 8C–E) revealed that the inactive and active state switching of *Pb*HARP is mainly mediated by the rearrangement of the HB domain, including rotation of α7 and shifting of the N-terminus of α8. Due to the structural rearrangement, α8 is split into two helices (termed α8a and α8b) in the complex. α8a and α8b are linked by one short loop, starting at the hinge residue E147 and ending at the catalytic residue D151 (Fig. 3D).

**Proper dimerization is essential for the function of *Pb*HARP.** *Pb*HARP exists as dimer in both the apo- and pre-tRNA$^{His}$-complexed structures. The dimerization interactions mediated by β4 (Fig. 1F, G) are maintained in both structures; however, due to the rearrangement of the HB domain, the dimerization interactions mediated by α7 and α8 varied. In the apo-form structure, L103, I104, V107 and I111 of α7 and L145 of α8 all locate around the central axis of HB, forming extensive hydrophobic interactions (Fig. 4A). V107 of the two partner molecules directly interact with each other, whereas L103 and I111 interact with I104 and L145 of the partner molecule, respectively. In the tRNA-complexed structure (Fig. 4B), L103 of the two *Pb*HARP monomers interact with each other. Stable hydrophobic interactions are also observed between the two V107 residues. Compared to the apo-form structure, the averaged distance between I111 of the two monomers is increased by 2 Å, altering the interactions between I111 and L145 in the complex.

In the apo-form structure, the N-termini of α8 helices are close to each other (Fig. 1C). However, the C-termini of α7 helices gathered together in the complex structure, allowing cooperative pre-tRNA binding by R116, R123, R138 and R142 of the two monomers (Fig. 2B–E, Supplementary Fig. 5A, C). Like the four

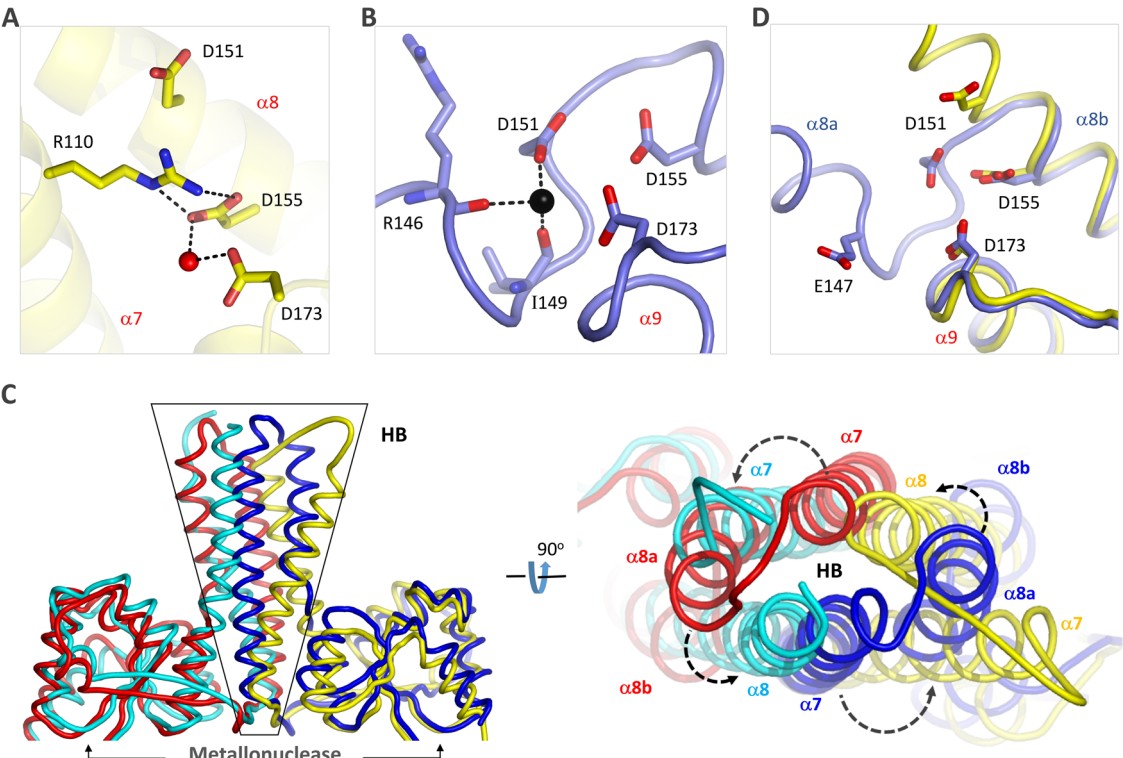

**Fig. 3 Conformational change of *Pb*HARP. A** Conformation of the Asp residues observed in the apo-form *Pb*HARP structure. **B** Conformation and cation coordination of the Asp residues observed in the pre-tRNA-complexed structure. $Ca^{2+}$ ion is shown as black sphere, the $Ca^{2+}$-coordinating residues are shown as sticks. **C, D** Structural superposition showing the conformational changes of *Pb*HARP. The apo-form structure is colored in yellow and cyan, whereas the pre-tRNA-complexed structure is colored in blue and red.

Arg residues, A118 is highly conserved (Supplementary Fig. 1). In the complex structure, the A118 residues center around the HB domain central axis, the distance between their side chains is only 3.6 Å (Fig. 4C). Except A118, many other hydrophobic residues, including A121 and Y141, are also arranged around the central axis (Fig. 4C).

To investigate the functional importance of HARP dimerization, we constructed and purified one *Pb*HARP mutant, L103E/V107E, in which the hydrophobic L103 and V107 are simultaneously substituted by negatively charged Glu residues. Different from WT *Pb*HARP, the size exclusion chromatographic study indicated that the L103E/V107E mutant mainly exists as monomer with an elution volume of 15.6 mL (Fig. 4D). Compared to WT *Pb*HARP, the L103E/V107E mutant is less stable, indicated by its lower melting temperature (51.1 °C) measured by the Nano-DSF method (Supplementary Fig. 9A, B). In addition, we also performed CD spectra analysis for the two *Pb*HARP proteins (Supplementary Fig. 10). In consistent with the crystal structure (Supplementary Fig. 4A, B), the CD results (Supplementary Fig. 10A, B) showed that WT *Pb*HARP possesses ~50% α-helices and 10% β-strands (Parallel + Antiparalel) at 20 °C; the calculated Tm value is ~70.0 °C. Different from the WT protein, the L103E/V107E mutant (Supplementary Fig. 10C, D) possesses ~30% α-helices and 20% β-strands (Parallel + Antiparalel) at 20 °C; the Tm value of the mutant is about 53.0 °C. These results suggested that the L103E/V107E mutation affected the proper folding of *Pb*HARP, which may play important role in the lower stability of the L103E/V107E mutant.

In addition to L103E/V107E, we also constructed the A118E/A121E mutant of *Pb*HARP, in which A118 and A121 are substituted by Glu. The elution volume (14.4 mL) of A118E/A121E is similar to that of WT *Pb*HARP, suggesting that it also exists as dimer (Fig. 4D). However, different from WT *Pb*HARP, the pre-tRNA binding affinity of A118E/A121E is weak; no pre-tRNA binding at all was observed for the L103E/V107E mutant (Supplementary Fig. 11A). WT *Pb*HARP can cleave pre-tRNA, but detectable pre-tRNA$^{His}$ cleavage activity was neither observed for the L103E/V107E nor for the A118E/A121E mutant under identical reaction conditions (Fig. 4E), probably due to their low stability and/or difficulty to form a functional dimer.

## Length of the acceptor stem affects pre-tRNA cleavage by HARPs

Compared to pre-tRNA$^{Gly}$, pre-tRNA$^{Cys}$ and pre-tRNA$^{Phe}$, the pre-tRNA$^{His}$ cleavage activity of *Pb*HARP is significantly weaker (Fig. 1B, Supplementary Fig. 3B). In the structure of the complex, *Pb*HARP mainly recognizes the shape and backbone of the TψC-loop (Fig. 2B–E), a structural element conserved in all tRNAs. However, compared to other pre-tRNAs (Supplementary Fig. 3A), the acceptor stem of pre-tRNA$^{His}$ is one base pair longer, due to the pairing of G-1 and C72 which is not visible in Supplementary Fig. 5B. During pre-tRNA 5'-end maturation, the phosphodiester bond between nucleotides -1 and +1 is usually cleaved. To investigate whether the extra base pair contributes to the weaker pre-tRNA$^{His}$ cleavage activity of *Pb*HARP, we transcribed three pre-tRNA$^{His}$ variants (Fig. 5A), His-7bp, His-6bp and His-9bp, which have seven, six and nine base pairs in the acceptor stem region, respectively.

Using 0.05 μM *Pb*HARP and 1 μM native or mutated pre-tRNA$^{His}$, we performed in vitro cleavage assays. As depicted in Fig. 5B, no His-9bp cleavage activity was observed for *Pb*HARP. Although it can be cleaved, the cleavage efficiency of native pre-tRNA$^{His}$ was much weaker than those of His-7bp and His-6bp. In fact, the cleavage efficiency of His-7bp was comparable to those of pre-tRNA$^{Cys}$ and pre-tRNA$^{Phe}$ under identical reaction conditions (Supplementary Fig. 3B). Besides *Pb*HARP, we also performed in vitro cleavage assay using 0.05 μM *Tr*HARP

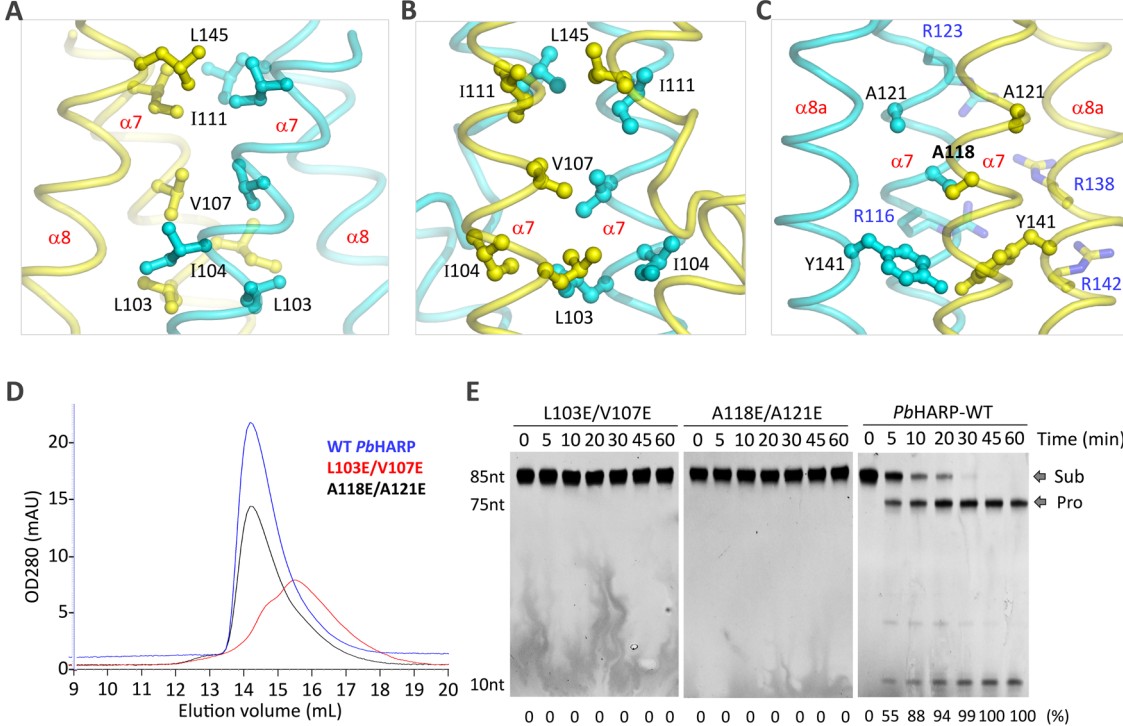

**Fig. 4 Verification of the functional importance of *Pb*HARP dimerization. A** Interactions involved in *Pb*HARP dimerization in the apo-form structure. **B, C** Interactions involved in *Pb*HARP dimerization in the pre-tRNA-complexed structure. **D** Size-exclusion chromatographic analysis of wild-type, L103E/V107E and A118E/A121E mutants of *Pb*HARP. **E** In vitro pre-tRNA^His cleavage assays catalyzed by WT *Pb*HARP, L103E/V107E and A118E/A121E mutants. The substrate cleavage percentage (%) is shown at the bottom of the gels. Experiments were repeated independently three times with similar results. Source data are provided in Source Data file.

(Fig. 5C). Similar to *Pb*HARP, *Tr*HARP failed to cleave His-9bp and also showed a clear preference for His-7bp.

**Oligomerization and assembly of *Tr*HARP**. Both *Tr*HARP and *Pb*HARP preferably cleaved the His-7bp variant (Fig. 5B, C), but their oligomerization states are different. *Tr*HARP eluted from the size exclusion column at ~11.4 mL in all buffers containing 100, 200 or 500 mM salt (Fig. 6A). A previous study suggested the possibility that Aq_880 may form either six trimers or three hexamers[30] (inferred from size exclusion chromatography profiles), but both *Pb*HARP and *Tc*HARP formed dimers in the crystal structures (Fig. 1C–F). To gain more insights into the assembly of HARPs, *Tr*HARP collected from the peak of the size-exclusion columns was subjected to electron microscopy. As depicted in Fig. 6B and S12A, *Tr*HARP produced many well-defined particles on the negative stain CCD images, which can be divided into 20 different classes (Fig. 6C).

The 2D class average showed an open ring-like conformation of *Tr*HARP. The shape and size of the ring matched well with an assembly of six *Pb*HARP dimers that arrange in a left-handed orientation (Fig. 6D). Oligomerization of *Tr*HARP is mainly mediated by the metallonuclease domains, which form the main body of the ring; the HB domains of *Tr*HARP all point toward the outside and are roughly vertical to the central rotation axis of the ring. Dimer_1 and Dimer_6 reside near the cleft of the ring, the rotation angle between them is ~80° (Supplementary Fig. 12B). The rotation angles are ~60°–70° between Dimer_4 and the two neighboring dimers (Dimer_3 and Dimer_5). The rotation angles are ~50° between Dimer_1 and Dimer_2, Dimer_2 and Dimer_3, and Dimer_5 and Dimer_6. On the average, the rotation angle between all the interacting dimers is ~55°.

**HARP binds and cleaves pre-tRNA in a cooperative mode**. In addition to the AB dimer that interacts with pre-tRNA^His, there is one more *Pb*HARP dimer (the CD dimer) within the asymmetric unit of the *Pb*HARP/pre-tRNA^His complex. Possibly due to crystal packing, no pre-tRNA^His is found near the CD dimer. Similar to *Tr*HARP, the AB and CD dimers of *Pb*HARP interact with each other via their metallonuclease domains; the rotation angle between the two dimers is ~55° (Supplementary Fig. 12C), which well matched with the averaged angle observed in the *Tr*HARP model (Supplementary Fig. 12B). *Pb*HARP appeared as dimer during the size-exclusion chromatographic analysis, but the complex structure and its similarity with *Tr*HARP suggested that *Pb*HARP has the potential to form larger oligomers under proper conditions. The incorporation of additional *Pb*HARP dimers can be easily modeled by rotating the complex structure along the 2-fold axis of either the AB dimer (Fig. 7A) or the CD dimer.

Based on pre-tRNA^His, the original AB dimer and the predicted EF dimer, one plausible pre-tRNA binding and cleavage model can be produced (Fig. 7B). In the model, the AB dimer is responsible for pre-tRNA binding, whereas the processing is performed by the EF dimer. The model can well explain the substrate preference of *Pb*HARP (Fig. 5B). For His-7bp, the acceptor stem is composed of 7 base pairs. When the elbow region of His-7bp is recognized by *Pb*HARP AB dimer, the first base pair of the acceptor stem, G1:C71, reaches the catalytic site of molecule E of the EF dimer. The phosphate backbone of G1 can readily coordinate with the catalytic cations, allowing the cleavage reaction to occur. G-1 pairs with C72 in native pre-tRNA^His, and such pairing may interfere with the conformational change and metal ion coordination at the scissile phosphodiester bond between G-1 and G1. The acceptor stem of His-9bp is 2 bp longer than that of His-7bp; the longer acceptor stem likely

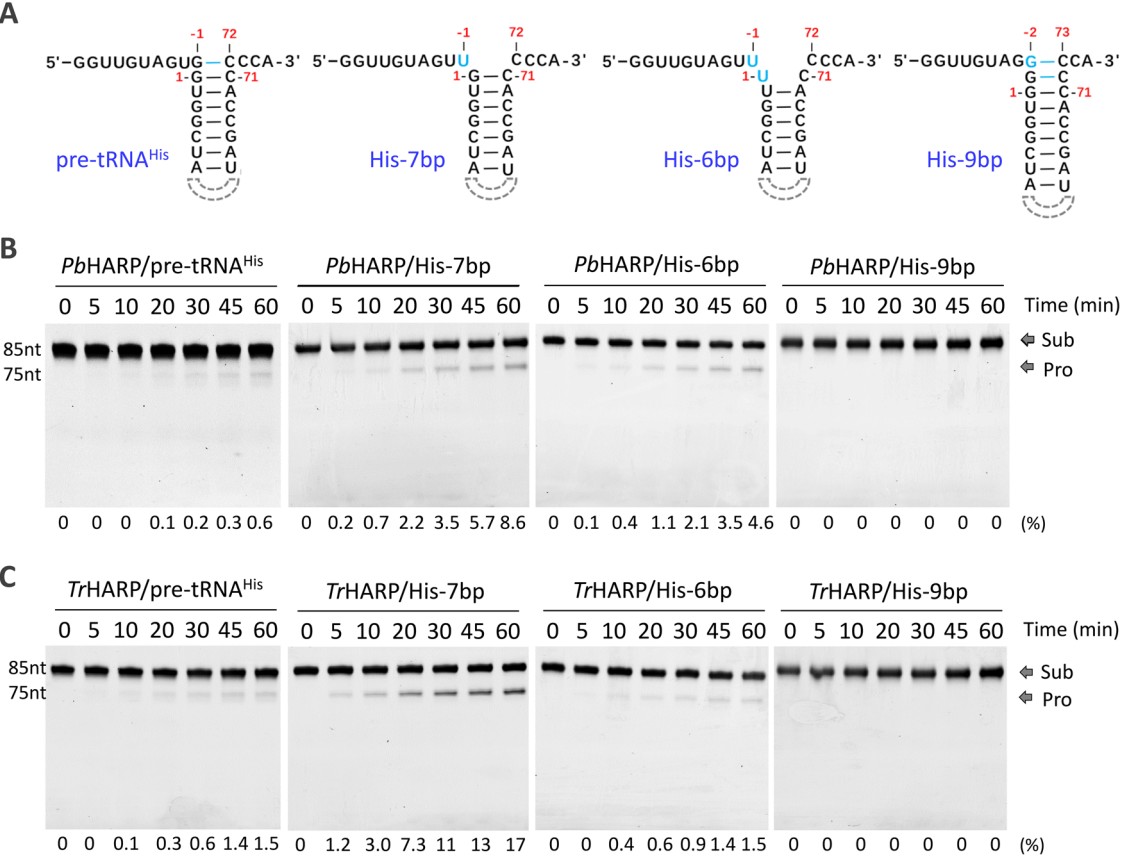

**Fig. 5 Pre-tRNA<sup>His</sup> mutation and cleavage by HARP. A** Sequence and secondary structure of 5'-end, acceptor stem and 3'-end of the native and mutated pre-tRNA<sup>His</sup>. **B** In vitro cleavage assays using 1 μM pre-tRNA and 0.05 μM *Pb*HARP. **C** In vitro cleavage assays using 1 μM pre-tRNA and 0.05 μM *Tr*HARP. The substrate cleavage percentage (%) is shown at the bottom of the gels. Experiments were repeated independently three times (**B**, **C**) with similar results. Source data are provided in Source Data file.

prevents His-9bp from simultaneously interacting with the AB and EF dimers.

Encouraged by the consistence between the in vitro cleavage assay results (Fig. 5B, C) and the pre-tRNA binding and cleavage model (Fig. 7B), we did further sequence and structural analysis. Besides the four Arg residues involved in pre-tRNA elbow recognition (Fig. 2B–E), two other Arg residues are also highly conserved in HARPs, corresponding to R108 and R146 in *Pb*HARP (Supplementary Fig. 1B). In the proposed pre-tRNA binding and cleavage model, the side chains of R146 of molecule E and R108 of molecule F are very close to the phosphate backbones of G-1 and G1 (Fig. 7C). To investigate the potential function of these Arg residues, we constructed two *Pb*HARP mutants, R108A and R146A. Although Ala substitution of R108 and R146 had no strong impact on pre-tRNA binding by *Pb*HARP (Supplementary Fig. 11B), no detectable pre-tRNA<sup>His</sup> cleavage activity was observed for the R108A or the R146A mutant (Fig. 7D).

Pre-tRNA<sup>His</sup> is recognized by the *Pb*HARP AB dimer in the complex, but neither R108 nor R146 of the AB dimer is involved in the interaction. These structural observations combined with the inactivity of these mutants further suggested that *Pb*HARP binds and cleaves pre-tRNA in a cooperative mode. To better demonstrate the cooperation between *Pb*HARP dimers, we performed in vitro crosslinking assays using WT *Pb*HARP and His-7bp (Supplementary Fig. 13). In addition to the monomer, *Pb*HARP dimers were also observed on the SDS-PAGE gel upon the addition of suberic acid bis (3-sulfo-N-hydroxysuccinimide ester) sodium salt (BS3), a chemical reagent that can crosslink the

side chains of two lysine residues. Although not clearly detectable in the absence of His-7bp, bands tentatively assigned to the *Pb*HARP tetramer (complex of two *Pb*HARP dimers) were clearly enhanced in the presence of His-7bp.

## Discussion

Here we present results of structural and biochemical analysis of HARPs from four organisms, *P. bacterium*, *T. celer*, *T. minervae* and *T. ruber*. Like Aq_880 from *A. aeolicus*[30], in vitro cleavage assays showed that *Pb*HARP, *Tc*HARP, *Tm*HARP and *Tr*HARP can all remove the 5'-end of pre-tRNA, suggesting that pre-tRNA 5'-end processing is one conserved activity of HARP. HARP and the eukaryotic PRORP are all members of the PIN domain superfamily[32]. However, unlike PRORP enzymes constituting the PRORP subfamily, HARPs belong to the PIN_5 cluster subgroup. The metallonuclease domain is shared by all PIN superfamily members. In the reported *At*PRORP1[34] and *Hs*MRPP3[39] structures, the metallonuclease domains adopt one well conserved conformation (Supplementary Fig. 14A, B). Similar to all PIN superfamily members, the metallonuclease domain of HARP is of α/β fold in nature, but only two central β strands (β1 and β4) and one α helix (α8b) can be superimposed with the corresponding regions in the eukaryotic PRORP structure (Supplementary Fig. 14C, D).

The activities of HARP and eukaryotic PRORP proteins are all cation-dependent. The apo-form *Hs*MRPP3 structure is in an inactive form, the active center is blocked by one of its own Arg residues[39]. Activation of *Hs*MRPP3 requires the help of the two

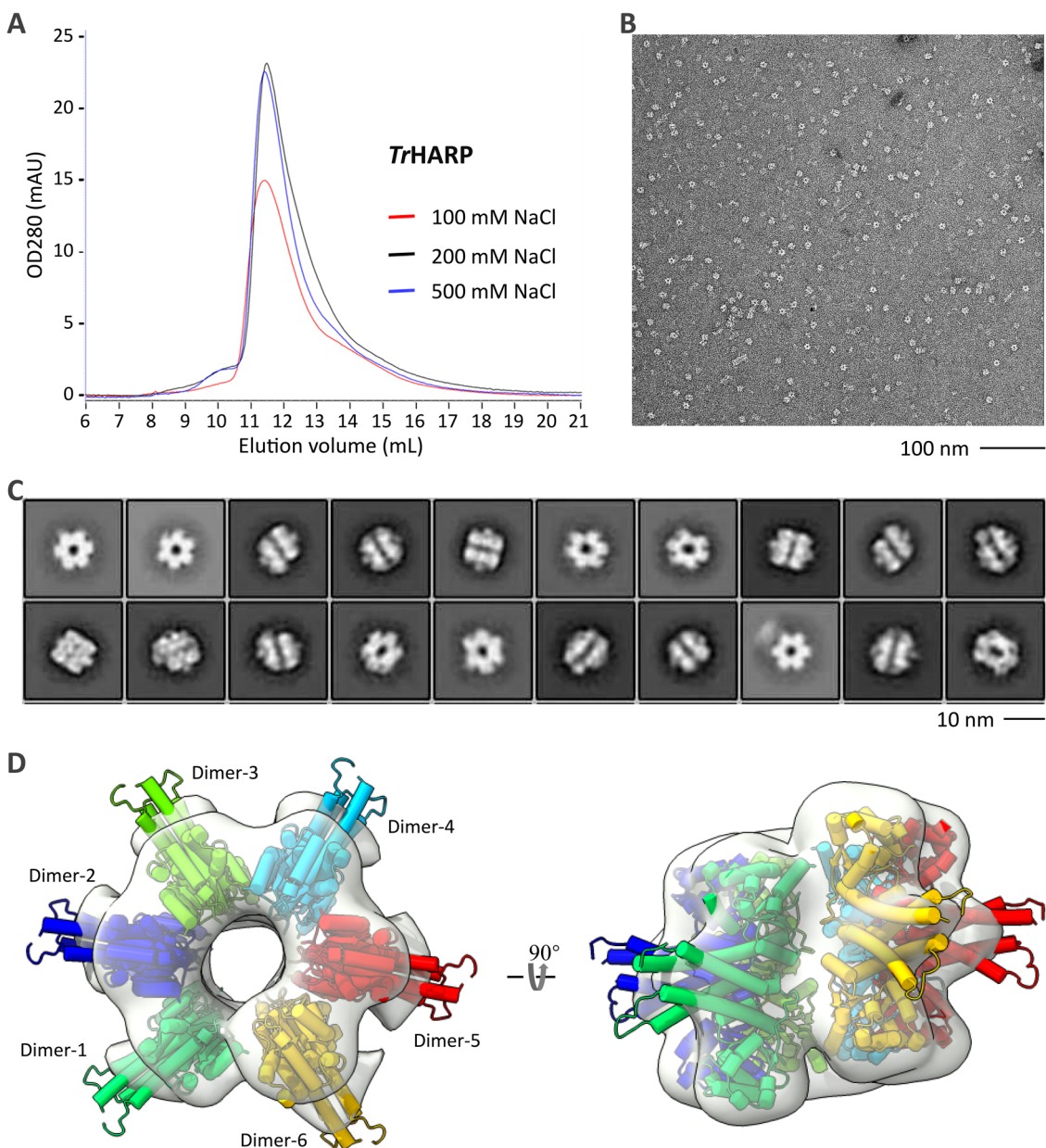

**Fig. 6 Electron microscopy analysis of *Tr*HARP. A** Size-exclusion chromatographic analysis of *Tr*HARP. Source data are provided in Source Data file. **B** A typical negative stain CCD image of *Tr*HARP. Experiments were repeated independently twice with similar results. **C** The reference-free two-dimensional class averages of *Tr*HARP. **D** The electron microscopy density maps of *Tr*HARP, fitted with *Pb*HARP dimer.

partner proteins, *Hs*MRPP1 and *Hs*MRPP2. The apo-form structures of *Pb*HARP and *Tc*HARP are also in an inactive form. However, in vitro cleavage assays confirmed that all HARPs are active on their own (Fig. 1B). Structural comparison of the apo- and the pre-tRNA-complexed *Pb*HARP structure suggested that HARP undergo self-activation via a large conformational change to the catalytically active form (Fig. 3B–D). When the catalytic *Pb*HARP and *At*PRORP1 structures are superimposed based on their central β strands and α helix, D155 and D173 of *Pb*HARP can align well with D475 and D493 of *At*PRORP1, respectively. However, the third catalytic Asp residue (D151) of *Pb*HARP is positioned in mirror image to D399 of *At*PRORP1 (Supplementary Fig. 14E). This mirror-image position of D151/ D399 and the self-activation ability distinguish HARP from eukaryotic PRORPs.

As observed in their crystal structures, both *At*PRORP1 and *Hs*MRPP3 contain an N-terminal PPR domain. *At*PRORP1 PPR is composed of five intandem PPR motifs; it is one of the typical proteins with multiple PPR motifs. In the complex structure of *Pb*HARP, the pre-tRNA<sup>His</sup> is recognized by the HB domain. Similar to the typical PPR motif, the HB domain is mainly composed of α–helices. Dimerization of HARP brings two HB domains together, structurally mimicking two PPR motifs arranged in tandem. Comparison with the recently reported *At*PRORP1 PPR/tRNA<sup>Phe</sup> complex[40] showed that the HB domain of HARP and *At*PRORP1 PPR both recognize the TψC-loop of pre-tRNA (Supplementary Fig. 15A, B), but the orientations of the HB domain and PPR domain are roughly vertical to each other (Supplementary Fig. 15C). Sequence and structure-based alignment indicated that the helices of the HARP HB and the *At*PRORP1 PPR domain are arranged in a reversed order (Supplementary Fig. 15D, E). However, one Arg residue is positioned at identical position in the two protein/tRNA complex structures (Supplementary Fig. 15F).

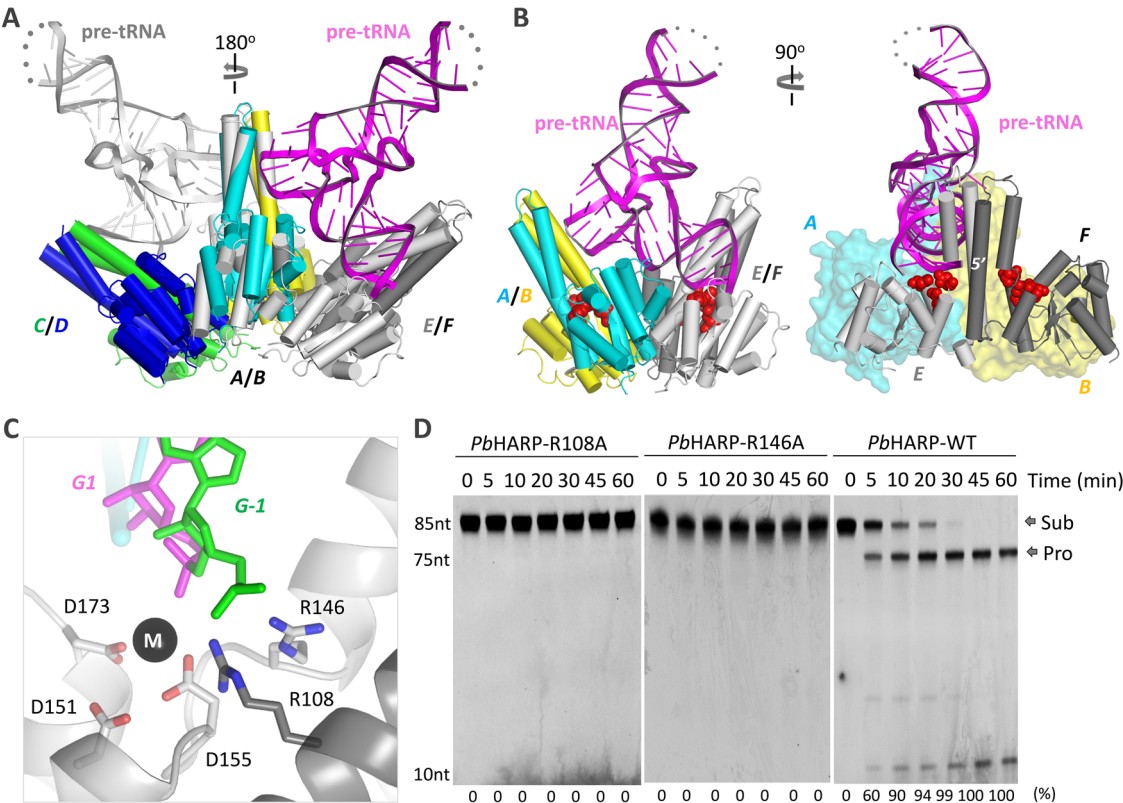

**Fig. 7 Model for pre-tRNA 5′-end processing by HARPs. A** Cartoon presentation of the original *Pb*HARP/pre-tRNA$^{His}$ complex and the one generated through rotation along the 2-fold axis of *Pb*HARP AB dimer. **B** Proposed pre-tRNA 5′-end processing model. Pre-tRNA and the AB dimer of *Pb*HARP are directly adopted from the complex, the EF dimer of *Pb*HARP is generated through rotation of the structure. **C** Conformation of cation, cleavage site nucleotides, catalytic Asp residues, R108 and R146 in the proposed model. **D** In vitro pre-tRNA$^{His}$ cleavage assays catalyzed by R108A and R146A mutants of *Pb*HARP. The protein and pre-tRNA concentration are all fixed at 1.0 µM. The substrate cleavage percentage (%) is shown at the bottom of the gels. Experiments were repeated independently three times with similar results. Source data are provided in Source Data file.

The apo-form *At*PRORP1 structure adopts a V-shaped conformation. The two arms are formed by the metallonuclease domain and the PPR domain, linked by the central domain at the bottom. Although it has not been experimentally verified, it is likely that the V-shaped conformation is important for the function of *At*PRORP1. With such arrangement, once pre-tRNA is recognized by the PPR domain, the acceptor stem and 5′-end of pre-tRNA will be placed near the active site of the metallonuclease domain. Neither a single monomer nor dimer of HARP is sufficient for binding and cleavage of pre-tRNA substrate, but the structure, mutagenesis and crosslinking assay results suggest that two dimers should be sufficient for the function of HARP. Interestingly, although not identical to *At*PRORP1, the two HARP dimers also seem to adopt a V-shaped conformation (Fig. 7B).

The total size of HARP is only about 1/3 of *At*PRORP1 and *Hs*MRPP3, the latter even requiring two additional protein cofactors (*Hs*MRPP1 and *Hs*MRPP2) for in vivo activity. However, HARP performs the same reaction as *At*PRORP1 and the *Hs*MRPP complex. A similar phenomenon has also been observed for some other RNA processing enzymes, such as RNase III and Trax. Bacterial RNase III is around 200 amino acids in length and functions as a homodimer. The human RNase III homology protein, Dicer, is about 2000 amino acids in length. Although different in size and domain architecture, bacterial RNase III and human Dicer catalyze the identical dsRNA cleavage reaction[35]. In the RNAi pathway in human and *Drosophila*, the nicked RNA passenger strand is degraded by C3PO, the complex of Trax and Translin[41,42]. Via oligomerization, Trax alone is

capable of RNA degradation in *Nanoarchaeum equitans*[43]. Very recently, the cryo-EM structures of Aq_880 and HARP from *Halorhodospira halophila SL1* (Hh) were reported[44,45]. Similar to *Tr*HARP, Aq_880 and *Hh*HARP assembled into dodecameric structures (Fig. 6D and Supplementary Fig. 16A, B). Structural comparison revealed that the overall folding of the individual *Hh*HARP dimer is similar to that of *Pb*HARP in the apo-form structure (Supplementary Fig. 16C), whereas the folding of Aq_880 dimer is more close to *Pb*HARP in the complex structure (Supplementary Fig. 16D). Superposition of Aq_880 and the *Pb*HARP/pre-tRNA$^{His}$ complex structures further confirmed the cooperative pre-tRNA binding and cleavage mode of HARP (Supplementary Fig. 16E). Both our and the reported structures of HARP support the idea that oligomerization and division of function between mono- or dimers may be a more common way (than previously thought) to provide small enzymes with substrate binding and catalytic abilities.

RNase P exists in two distinct forms (PRORP and the RNA-based RNase P) and has been considered the most excellent example for the evolutionary transition theory[31]. Owing to their functional importance, the structure and catalytic mechanism of the RNA-based RNase P have been extensively studied[46–50]. The existence of PRORP was proposed over 30 years ago[51,52], but the identification and characterization of PRORP were mainly reported in recent years[28,29,53]. HARP is not essential for Archaea and Bacteria that also encode an RNA-based RNase P, whereas it is the only RNase P expressed in *A. aeolicus* and many related *Aquificaceae*, indicating their important role in tRNA maturation. Compared to PRORPs from animal mitochondria, plant

chloroplast, and trypanosomal mitochondria, very little is known about HARP from Archaeal and Bacteria. The *Pb*HARP and *Tc*HARP structures we report here are the only available crystal structures of HARP. Since no metallonuclease domain is present in the *At*PRORP1 PPR/tRNA^Phe complex[40], the *Pb*HARP/pre-tRNA^His structure represents the only tRNA-complexed structure of a full-length protein-only RNase P. Instead of being dodecamers, *Pb*HARP and *Tc*HARP exist as dimer in the structures (Fig. 1C, E). In vitro size-exclusion chromatography showed that the oligomerization state of *Tc*HARP is changeable (Supplementary Fig. 2B). We believe that a tetramer is the minimal catalytic unit of HARP in vivo, whereas dodecamerization may allow HARP to bind and cut multiple pre-tRNAs simultaneously, as proposed recently[44,45]. Our study reveals the molecular basis for substrate binding and cleavage by HARP proteins.

## Methods

**Construction of recombinant plasmids.** All genes that contain codon-optimized DNA sequences of Aq_880, *Pb*HARP, *Tc*HARP *Tm*HARP, *Tr*HARP and *E. coli* RnpA (Supplementary Table 2) were purchased from Shanghai Generay Biotech Co., Ltd, China. The target fragments were amplified by PCR, cleaved by BamHI (New England Biolabs, R3136L) and XhoI (New England Biolabs, R0146L), and cloned into pET28a-Sumo vectors. All recombinant proteins contain one 6 × His-Sumo tag at their N-termini. The recombinant plasmids of *Pb*HARP mutants were constructed by overlap PCR using the wild-type (WT) *Pb*HARP plasmid as template and primers listed in Supplementary Table 3. For transcription of *E. coli* RNase P RNA, the *rnpB* gene with T7 promoter and hammerhead ribozyme (Supplementary Table 2) was inserted into a pUC18 vector at the HindIII and EcoRI restriction sites. Sequences of all plasmids were confirmed by DNA sequencing.

**Expression and purification of recombinant proteins.** All HARP proteins were expressed and purified using similar procedures. The recombinant plasmids were transformed into *Escherichia coli* BL21(DE3) competent cells and cultured in Lysogeny broth (LB) medium supplemented with 50 μg/mL kanamycin at 37 °C. When the $OD_{600}$ reached 0.6, isopropyl β-D-1-thiogalactopyranoside (IPTG, final concentration of 0.2 mM, Sangon Biotech, A600168-0100) was added to induce target protein expression. For accumulation of recombinant proteins, the cultures were incubated at 18 °C for an additional 18 h. For expression of Seleno-methionine (Se-Met) substituted *Tc*HARP protein, cells were cultured in M9 medium supplemented with 30 mg/L L-selenomethionine (J&K Scientific, S0442).

Cells were collected by centrifugation, resuspended in Buffer A (20 mM Tris-HCl pH 8.0, 500 mM NaCl, 25 mM Imidazole), and lysed by ultrahigh-pressure homogenizer. Cellular debris was removed by centrifugation and the supernatant was loaded onto a HisTrap HP column (GE Healthcare). After washed by high salt buffer (20 mM Tris-HCl pH 8.0, 2 M NaCl), the bound protein was eluted with a linear gradient of buffer B (20 mM Tris-HCl pH 8.0, 500 mM NaCl, 500 mM imidazole). The proteins were treated with Ulp1 and dialyzed against buffer C (20 mM Tris-HCl pH 8.0, 500 mM NaCl). The samples were loaded onto the HisTrap HP column again, the flow-through was collected, concentrated, and applied to a Hiload 16/600 Superdex 200 column (GE Healthcare). For *E. coli* RnpA purification, the target protein eluted from HisTrap HP column was treated with Ulp1, diluted with 20 mM Tris-HCl pH 8.0 buffer and loaded onto a HiTrap Heparin column (GE Healthcare). The bound protein was eluted with a linear gradient of 20 mM Tris pH 8.0, 1 M NaCl buffer. The eluted protein was concentrated and applied to a Hiload 16/600 Superdex 75 column (GE Healthcare) equilibrated with 20 mM Tris pH 8.0, 200 mM NaCl buffer. Peak fractions containing the target proteins were pooled and concentrated. The purity of protein was analyzed by 15% SDS-PAGE gel, the concentration was measured using a UV-spectrophotometer at 280 nm.

**In vitro RNA transcription.** All pre-tRNAs and *E. coli* RNase P RNA were synthesized by in vitro transcription using the same method. The template that contains a T7 promoter sequence upstream of the pre-tRNA or RNase P RNA coding region was obtained by overlap PCR using the primers listed in Supplementary Table 4. In vitro transcription reaction was carried out by mixing T7 RNA polymerase (prepared in laboratory), NTPs and template together and incubated at 37 °C overnight. Next day, the DNA template was digested by RNase-free DNase I (TaKaRa, 2270 A) at 37 °C. To remove the hammerhead ribozyme and increase RNase P RNA yield, the DNase I-digested transcription sample was incubated overnight at 4 °C. All RNA samples were mixed with equal volume of formamide, heated at 95 °C for 10 min, and chilled on ice. The in vitro transcripts were then separated by 10% (for pre-tRNA) or 6% (for RNase P RNA) denaturing UREA-PAGE gels (60 W for 6 h) in 1× TBE buffer. The target pre-tRNA or RNase P RNA band was excised from the gel. The gel slice was eluted using elution buffer (300 mM sodium acetate pH 5.2, 1 mM EDTA pH 8.0) at 8 °C overnight. The RNA

was precipitated by mixing with 1/10 volume sodium acetate pH 5.2 and 1 volume of isopropanol and freezing at −80 °C. The sample was centrifuged at 13000 × g. for 30 min at 4 °C, the supernatant was discarded and the pellet was washed by 75% ethanol and air-dried. Finally, the RNA was dissolved in RNase-free water, quantified by absorbance, and stored at −80 °C. All purified pre-tRNAs were analyzed by 10% denatured UREA-PAGE gels and 6% native PAGE gels.

**In vitro RNA binding assays.** Electrophoretic mobility shift assays (EMSA) were used to analyze the pre-tRNA binding ability of wild type full-length, truncated and/or mutated *Pb*HARP proteins. Assays with a final volume of 10 μL were set up. The reaction mixtures that included 1 μM pre-tRNA and protein serially diluted from 1 μM to 10 μM (or 20 μM) were incubated on ice for 1 h in 20 mM Tris-HCl pH 8.0, 200 mM NaCl, 5% (v/v) glycerol buffer. After incubation, 2 μL 10 × DNA loading buffer (Vazyme) was added. Samples were loaded onto a precooled 1% agarose gel and then separated by electrophoresis (100 V for 20 min at 4 °C) in 1 × TAE buffer. The gel was stained by Gel Red and imaged using gel scanner. Intensities of the substrate bands were quantified by ImageJ. The percentage of binding, for each protein concentration, was calculated. Data were then fitted to the equation $Y = B_{max} * X^h/(K_d^h + X^h)$ using nonlinear regression (curve fit) in GraphPad Prism. The dissociation constants ($K_d$) were determined from the regression curve.

**In vitro pre-tRNA cleavage assays.** *Pb*HARP and mutants were serially diluted to 10 μM (other HARPs were 0.5 μM) with processing buffer [20 mM Tris-HCl pH 9.0, 100 mM NaCl, 5% (v/v) glycerol] and pre-incubated at 37 °C for 5 min. The mixture of 33.75 μL processing buffer, 2.25 μL 100 mM $MgCl_2$, and 4.5 μL 10 μM pre-tRNA was also pre-incubated at 37 °C for 5 min. The reactions were initiated through addition of 4 μL diluted protein. The final concentrations of HARP, pre-tRNA, $MgCl_2$ are 1 μM (or 0.05 μM), 1 μM, 5 mM, respectively. At specific time points, 5 μL aliquots of the reaction were quenched with 5 μL 2 × Formamide loading buffer [90% (v/v) formamide deionized, 20 mM EDTA, 1% SDS, 0.02% bromophenol blue, 0.02% xylene cyanol] and heated at 95 °C for 5 min. Samples were centrifuged and separated by 10% denaturing UREA-PAGE gels (10 W for 2 h) in 1× TBE buffer. The bands were stained by Gel Red and visualized by Typhoon FLA 9000 Imaging Scanner.

For cleavage site specificity assay, *E. coli* RNase P holoenzyme was prepared by mixing RnpA and RNase P RNA with a 1:1 molar ratio in processing buffer. The mixture was incubated at 4 °C for 10 min. RNase P (*E. coli* RNase P holoenzyme, *Pb*HARP or Aq_880) and the reaction mixtures (15 μL processing buffer, 1 μL 100 mM $MgCl_2$ and 2 μL 50 μM pre-tRNA^Phe) were incubated at 37 °C for 5 min. The reactions were initiated through addition of 2 μL diluted RNase P. The final concentrations of RNase P, pre-tRNA^Phe and $MgCl_2$ were 0.25 μM, 5 μM and 5 mM, respectively. The reactions were terminated after 15 min by heating to 95 °C for 5 min. Samples were centrifuged and separated by 20% denaturing UREA-PAGE gels (10 W for 2 h 25 min) in 1× TBE buffer.

**In vitro chemical crosslinking assays.** The buffer of *Pb*HARP was exchanged with 20 mM HEPES pH 7.9, 500 mM NaCl through protein concentration in an ultrafiltration tube. For crosslinking of complex, *Pb*HARP (the final concentration is 1 mg/ml) and His-7bp were mixed with 10 mM $CaCl_2$ in gel filtration buffer at a molar ratio of 1:1.2. For cross-linking of apo-*Pb*HARP, *Pb*HARP was mixed with corresponding volume of water in gel filtration buffer. The mixtures were incubated at 4 °C for 40 min and divided into 10 μL per tube. Chemical crosslinking experiments were started by addition of BS3 (ThermoFisher Scientific, 21580) reagent at a final concentration of 0 mM, 0.01 mM, 0.03 mM, respectively. Samples were incubated at 25 °C for 1 h. In the crosslinking system, the final concentration of *Pb*HARP and RNA is 37.6 μM and 45.1 μM, respectively. To terminate the crosslinking reactions, 20 mM Glycine was added and then incubated at 25 °C for 15 min. Finally, 3 μL of 5×SDS PAGE loading buffer [250 mM Tris-HCl, pH 6.8, 50% (v/v) glycerol, 10% (w/v) SDS, 0.05% bromophenol blue, 5% (v/v) β-mercaptoethanol] was added into each tube and 5 μL samples were loaded to a 12% SDS-PAGE (300 V for 30 min) for analysis.

**Size-exclusion chromatography and electron micrographic analysis.** To analyze the oligomerization state of HARPs, 500 μL HARP proteins (300 μg) were applied to Superdex 200 Increase 10/300 GL column (GE Healthcare) equilibrated with buffers containing 20 mM Tris-HCl pH 8.0 and 100–500 mM NaCl. The flow rate was set at 0.4 mL/min. Besides HARP proteins, we also performed size-exclusion chromatography analysis for 100 μL standard marker proteins using the same column equilibrated with 20 mM Tris-HCl pH 7.0 and 150 mM NaCl buffer. To further determine the oligomerization state of *Tr*HARP, 400 μg protein was applied to Superdex 200 Increase 10/300 GL column equilibrated with 20 mM Tris-HCl pH 8.0 and 200 mM NaCl. The peak fraction was diluted to 40 μg/mL using the same buffer. 5 μL sample was absorbed onto the glow-discharged carbon-coated copper grids (200 mesh) purchased from Beijing Zhongjingkeyi Technology Co., Ltd, China and incubated for 1 min, the excess solution was blotted off. After washing, the sample on the grid was negatively stained with 0.75% uranyl formate for 3.5 min and then air-dried. Finally, the grid was viewed with the Talos L120C transmission electron microscope (ThermoFisher Scientific) equipped with a

4 K × 4 K CETA CCD camera (FEI) at an accelerating voltage of 120 kV. Data were collected and images were recorded at a nominal magnification of 92 000×, corresponding to a pixel size of 1.55 Å. The contrast-transfer function (CTF) parameters of each micrograph were determined using CTFFIND4. Single particles were picked and processed using RELION3.0[54]. 29647 particles were picked by using LoG-based autopicking from RELION3.0. Reference-free 2D classification was generated using RELION. 2D class-averages were used to generate de novo 3D model and 3D classification by RELION.

**Nano-differential scanning fluorimetry (nano-DSF) analysis**. Nano-DSF experiments were performed using a Prometheus NT.48 instrument from Nano-Temper Technologies (Munich, Germany). 20 μL of 1.0 mg/mL WT *Pb*HARP and the L103E/V107E mutant samples were prepared. Each sample was dissolved in 20 mM Tris-HCl pH 8.0, 500 mM NaCl, 5% (v/v) glycerol buffer. For each analysis, 10 μL samples were loaded into PR NT.48 standard capillaries from NanoTemper Technologies. Samples were subjected to a temperature gradient of 1 °C/min from 20 °C to 90 °C and the fluorescence was constantly monitored. Protein unfolding was detected by following the change in tryptophan fluorescence at emission wavelengths of 330 and 350 nm. The ratio between the emission intensities at 350 nm and 330 nm was used to track the structural changes with increasing temperature. Tm value was determined from the first derivative nano-DSF curve.

**Circular dichroism (CD) spectra analysis**. Both WT *Pb*HARP and the L103E/V107E mutant were dissolved in 5 mM Tris pH 8.0, 500 mM NaF buffer with a final concentration 0.2 mg/mL. 300 μL sample or buffer was loaded into a quartz cuvette of 1 mm path length (Hellma, Germany). The CD spectra was recorded on a nitrogen-flushed Chirascan CD spectrometer (Applied Photophysics, UK). The wavelength range of 190–260 nm was recorded. The step and band width were set at 1 nm. The scanning time per point was 0.5 s. For temperature control, the heating rate was set at 1 °C/min in a range of 20 °C–80 °C (with a temperature interval measurement of 2 °C). Secondary structure content (%) was calculated by CDNN software. Tm value was determined using nonlinear regression (curve fit) in GraphPad Prism software.

**Crystallization and data collection**. For crystallization of apo-form HARP, 10 mg/ml protein samples were used. For crystallization of HARP/pre-tRNA complex, HARP was mixed with pre-tRNA and 10 mM CaCl₂ in gel filtration buffer, the molar ratio between HARP and pre-tRNA is 1:1.2. The mixture was incubated at 4 °C for 1 h. The initial crystallization conditions were identified at 16 °C using the Gryphon crystallization robot system and commercial crystallization kits. The sitting-drop vapor diffusion method with the 3-drop intelliplate plates were utilized during both initial screening and optimization process. The crystallization condition for *Tc*HARP and Se-*Tc*HARP was 30% (w/v) PEG 3000, 100 mM Tris-HCl pH 7.0, 200 mM NaCl. The crystals of apo-form *Pb*HARP grew in the buffer composed of 2000 mM Ammonium sulfate, 100 mM Sodium citrate/Citric acid pH 5.5, whereas the crystallization conditions are composed of 8% (v/v) Tacsimate pH 8.0, 20% (w/v) PEG 3350 for the *Pb*HARP/pre-tRNA^His crystals.

All crystals were cryoprotected in reservoir solution supplemented with 25% (v/v) glycerol and snap-frozen in liquid nitrogen. The X-ray diffraction data were collected on beamlines BL17U and BL19U at the Shanghai Synchrotron Radiation Facility (SSRF). HKL3000 program[55] was used to process the data. The data collection and processing statistics are summarized in Supplementary Table 1.

**Structure determination and refinement**. The apo-form Se-Met substituted *Tc*HARP structure was solved by the single-wavelength anomalous diffraction (SAD) method with the Autosol program embedded in the Phenix suit[56]. The initial model was built using the Autobuilt program and then refined against the diffraction data using the Refmac5 program of the CCP4 suite[57]. The 2F₀ −F_c and F₀ −F_c electron density maps were regularly calculated and used as guide for the building of the missing amino acids using COOT[58]. The apo-form *Pb*HARP structure was solved by molecular replacement using the apo-*Tc*HARP structure as the search model with the phaser program of the CCP4 suite. The pre-tRNA^His-complexed *Pb*HARP structures was solved by molecular replacement using the apo-form *Pb*HARP structure as the search model. Nucleic acids, ions, water, and other molecules were all built manually using COOT. The complex structures were refined using the Refmac5 program of the CCP4 suite or the phenix.refine program of Phenix suit. The structural refinement statistics were summarized in Supplementary Table 1.

**Reporting summary**. Further information on research design is available in the Nature Research Reporting Summary linked to this article.

## Data availability
The data supporting the findings of this study are available from the corresponding authors upon reasonable request. Structural factors and coordinates have been deposited in the Protein Data Bank under accession codes 7E8J, 7E8K, and 7E8O for the apo-form *Tc*HARP, apo-form *Pb*HARP, and *Pb*HARP/pre-tRNA^His complex, respectively. Source data are provided with this paper.

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

## Acknowledgements

We thank the staff of BL17U1 beamline at the Shanghai Synchrotron Radiation Facility (SSRF), the staff of BL18U1, BL19U1 beamlines of National Faciltiy for Protein Science Shanghai for help during data collection. This work was supported by the National Natural Science Foundation of China (32171197 and 31870721 to J.G.) and the Key Research and Development Project of China (2016YFA0500600 to J.G.).

## Author contributions

Y.Y.L. performed the crystallization and biochemical studies with the help of H.H.L., X.C., Z.W.S., Y.X.Z., Q.Y.S., X.Z., J. Y. and C.L.C. Y.Y.L. and Y.Q.G. collected and processed the crystal diffraction data. S.C.S., Y.Y.L. and G.L.L. performed the electron microscopy study. Y.Y.L., J.Z.L., J.B.M, and J.H.G analyzed the data. Y.Y.L. and J.H.G. wrote the manuscript.

## Competing interests

The authors declare no competing interests.
