## [Peer Review File · Nature Communications]

Title: Crystal structures and insights into precursor tRNA
5'-end processing by prokaryotic minimal protein-only RNase
PREVIEWER COMMENTS

Reviewer #1 (Remarks to the Author):

Review NCOMMS-21-21769

The authors report two crystal structures of HARP dimers, one in complex with a pre-tRNA, which differ in the conformation of the domain that recognizes the tRNA elbow region. The structures provide insight into the architecture of HARP metallo-nucleases and tRNA recognition at the atomic level. Furthermore, a convincing model of cooperative substrate recognition by two neighboring dimers is proposed. The aforementioned aspects represent the strength of the manuscript. Several weaknesses are associated with the peripheral biochemical experiments (pre-tRNA processing experiments, size exclusion chromatography) as outlined below.

Major comments

1) line 84 ("Knockout of the HARP gene ..."), rewrite e.g.: "Knockout of the HARP gene showed no growth defects in *H. volcanii* and *M. mazei*, indicating that the RNA-based RNase P is the main and essential RNase P activity in these archaea 32."

Do not cite ref. 31 here; rather cite ref. 31 together with ref. 29 at the end of line 72.

2) Lines 138-141: the architectural features described here are not clearly recognizable in Fig. 1 or Fig. S4B. Please harmonize text and illustrations.

Line 144: the sentence " $\alpha 7$ is linked to the N-terminus of $\alpha 8$." Better say " $\alpha 7$ is separated from $\alpha 8$ by a short linker."

3) indicate $\alpha 7$ in Fig. 1G.

4) Line 176, change to: "... G-10 to U-2, U32 to A36 and C73/A74 located at the 5'-end, ..."

5) in Fig. S3B it is not clear if pre-tRNA^{His} is cleaved specifically at the canonical RNase P cleavage site or if this is non-specific degradation. Also, at the unusually high pH of 9 of the processing buffer, metal ion-induced degradation cannot be completely excluded. These aspects represent a general weakness of the presented 10% PAA gels that do not allow resolution/identification of specific cleavage products; appropriate would be 20-25% PAA gels to resolve the 5'-leader products, using, for example, *E. coli* RNase P or Aq_880 processing samples as reference.

6) Lines 263/264: are stable hydrophobic interactions between the V107 residues evident in both, the apo enzyme and the HARP-pre-tRNA complex structure?

- 7) Paragraph beginning in line 257: here the authors compare the interface between monomers in the dimer for the apo enzyme and the dimer-tRNA complex. The reader is here forced to switch back and forth between Fig. 1G and 4A. For easier reading, move Fig. 1G to Fig. 4 and only discuss the apo form interface together with the altered interface in the dimer-tRNA complex in the context of Fig. 4.
- 8) Line 280: the A118/121E mutant should be renamed to A118E/A121E. I find it surprising that this double mutant still dimerizes, as one expects disruption of the hydrophobic interface as well as electrostatic repulsion and steric clashes. Are you sure that the elution peak at 14.4 mL really represents dimers?
- 9) In general, the authors present size-exclusion chromatography profiles and claim that they represent monomers or certain oligomers (Fig. 4B, 6A, S2), but no marker proteins were run on the same column. Also, the methodological description of the size-exclusion chromatography (starting in line 554) should be completed, such that one can replicate the experiment.
- 10) Fig. 4C, 7D: Analysis of cleavage activity must include a positive control as part of the same experiment, to exclude that something else than the tested protein was responsible for inactivity.
- 11) Lines 295-303: this description of the pre-tRNA^{His} mutants is long-winded; I propose to shorten the text here by stating that variants with 7, 6 and 9 bp in the acceptor stem were constructed and to simply refer to Fig. 5A.
- 12) Lines 423-427, starting with "Sequence and ...": it is not immediately clear what the authors mean here by "arranged in a reversed order". It would be helpful to explain to the reader in more detail in the legend to Fig. S11 D and E what the idea is; also, mark the overlapping Arg residues (R212, R138) in Fig. S11E.
- 13) Line 456: with regard to structure and catalytic mechanism, ref. 47 is not appropriate here; I propose to cite here instead good reviews by Klemm et al. 2016, *Biomolecules*, doi: 10.3390/biom6020027, and Schencking et al. 2020 (https://doi.org/10.1007/978-3-030-57246-4_11), plus refs 44, 46 and Wan et al. 2019 (Cryo-electron microscopy structure of an archaeal ribonuclease P holoenzyme. *Nat Commun*, doi: 10.1038/s41467-019-10496-3).
- 14) Line 458: exchange current ref. 26 with "Holzmann et al. 2008, *Cell* (doi: 10.1016/j.cell.2008.09.013)"
- 15) Line 467/468: omit "... our study further supported the evolutionary transition theory." This study in no way addresses any aspects of RNase P evolution.
- 16) Legends to figures: some figure legends are extremely parsimonious, such as those in Fig. S2 and S6. At least a remark such as "For experimental details, see Methods section of the main text" would be appropriate.

Minor comments

- line 65: for *T. brucei* PRORP, cite Taschner et al. Cell Rep. 2012 (doi: 10.1016/j.celrep.2012.05.021)
- line 91: cite ref. 29 after "... with glucose in the medium."
- line 197, rewrite to e.g. "... we showed that PbHARP is also able to process pre-tRNACys and pre-tRNAPhe under identical reaction conditions (Fig. S3B)."
- line 199: "... is functionally relevant."
- line 209: "... cleavage activity at all was observed for the R138A mutant."
- line 223: "... D173A mutants (Fig. S6)."
- line 269: replace "incorporative" with "cooperative"
- line 279: "...15.6 mL (Fig. 4B)."
- line 294: pairing of G-1 and C72 is not visible in Fig. S5B.
- lines 311-313: shorten and combine the last two sentences: "Similar to PbHARP, TrHARP failed to cleave pre-tRNAHis-3 and also showed a clear preference for pre-tRNAHis-1." I also propose to rename the pre-tRNAHis variants to 'His 8bp', 'His 7bp' etc. for the sake of simplicity.
- line 332: change Fig. S7B to Fig. S8B
- lines 382-384 (last sentence before the discussion), rewrite: "Bands tentatively assigned to the PbHARP tetramer (complex of two PbHARP dimers) were clearly enhanced in the presence of pre-tRNAHis-1."
- line 387, change to: " Here we present results of structural and biochemical analyses of HARPs from four organisms, *P. bacterium*, *T. celer*, *T. minervae* and *T. ruber*."
- line 415: "... an N-terminal PPR domain."
- line 466, rephrase: "... pre-tRNA-complexed structure of a full-length protein-only RNase P

Reviewer #2 (Remarks to the Author):

The authors determined the first crystal structures of prokaryotic HARP enzymes with/without tRNA molecules. The structure revealed that HARP enzymes exist as a dimer mediated by two helical regions (HB region). The tRNA complexed structure and mutational analysis showed key basic residues of HB region are involved in tRNA recognition, particularly elbow recognition. The structures also revealed that HARP undergoes conformational changes in elbow recognition and active site formation. Furthermore, the authors demonstrated HARP assemble to the oligomer (hexamer of dimer: dodecamer) by electron microscopy analysis and suggested that the oligomerization is necessary to process pre-tRNA substrate. This study provides new insights into the field of RNase P research and unveils the structural basis of how the minimal protein-only RNase P recognizes its substrates. However, several discussion points and concerns need to be addressed in a revision of the manuscript.

Major issues

1. Title could be modified to convey the crystal structure of prokaryotic minimal protein-only RNase P. This article is not a review paper, but the current title is associated with all types of protein-only RNase P.

2. The authors need a more in-depth discussion of the oligomeric state of HARP.

- i) In which oligomeric state (tetramer, hexamer, or dodecamer?) do the authors think that HARP functions in vivo? The minimal catalytic unit seems to be a tetramer, but TrHARP assembles to the dodecamer.
- ii) What is the significance of dodecamer?
- iii) How many molecules of pre-tRNA can bind to the HARP dodecamer? The big picture would need to be discussed.
- iv) Is the low activity of PbHARP related to its difficulty in oligomerization? What is the difference in the structural basis between high and low activity HARPs? Difference of residues?
- v) In crystal packing, were there any interactions similar to the oligomeric state?
- vi) Very recently, other HARP dodecamer papers have been published, so please cite them for a better discussion.

3. Since the 5' leader sequence is not in the active site, the authors performed mutational analyses to eliminate the effect of crystal packing and to evaluate the important basic residues. However, the binding constants should be obtained for some representative residues instead of qualitative cleavage assays. With qualitative cleavage assays alone, readers cannot rigorously assess the importance of residues.

4. Since the structures of PbHARP and TcHARP are the first crystal structures of HARP family, the authors would need to describe the detail of the active site.

- i) The authors described only three residues (D151, D155, and D173) of the active site. Can authors propose other key residues from the structural information? Are there any other residues in the active site?
- ii) The author described the Mg ion essential for catalysis is coordinated with D151, D155, and D173, but NO rationale or previous research is provided in the manuscript.
- iii) In the structure of tRNA complex, Ca ions are bound to the active site. What is the binding position of the Ca ion compared to the Ca, Mg, and Mn ions in the previously determined crystal structure of *Arabidopsis thaliana* PRORPs?

5. The authors confirmed the oligomeric state of HARPs by size-exclusion chromatography. However, they have not shown the standard protein. Thus, the readers do not know how the peak positions of the dimer, oligomer, or monomer were determined.

- i) The authors would need to illustrate standard peak position and calibration curve.
- ii) Peak profile of L103E/V107E indicated a broad peak and low absorbance. The authors would need to check protein stability. The activity may be lost due to low stability.

Minor points

Ln24-25: Knowledge of substrate recognition and cleavage mechanism of eukaryotic PRORPs has been accumulated. It would be better to limit to HARP.

Ln59-67: It is a little difficult to understand the relationship between MRPP3 and PRORP.

Ln68: MRPP3-like PRORP brings up the image of PRORP that function with three subunits. Readers may mistakenly think that bacteria have Arabidopsis and Trypanosoma-type PRORP.

Ln83-84: The overall preservation of HARP does not appear to be very limited.

Ln90: lack of reference

Ln145: Please show b-factor distribution or weak electron density in supplemental figure

Ln161-162: Please describe the r.m.s.d.

Ln244: Could the tRNA-complex be described as the active state even though it is not active because Ca is bound to the active site?

Ln288-313: It might be difficult to follow the relationship between the name of the pre-tRNAHis variant and the number of Base pairs. Please give them names that are easy to understand.

Ln320: lack of reference

Ln320: Why didn't TcHARP check the oligomeric state by EM? A reason would be welcome.

Ln380: The authors should provide a brief description of the cross-linking details. Readers don't know what is being cross-linked.

Ln465-466: The authors described that the PbHARP/pre-tRNAHis structure represents the only pre-tRNA-complexed structure of full-length PRORP. However, the 5'leader sequence is not in the active site. it would be better to rephrase it

Figs: Please show the cleavage percentage at the bottom of the gel figures.

Fig.1: MRPP1/2 proteins might not be necessary in the main figure.

Fig.S7: Please illustrate Fo-Fc omit map of Ca ion. Model bias might be a concern.

Fig.S9: Please show sizes of protein markers.

Fig.Ss: The extra lines in the figure outline are seen.

We sincerely thank the reviewers for reading our manuscript with great care, we would also like to thank the reviewers for their very helpful comments, encouragements and criticisms as well. Based on their comments and suggestions, we have carefully revised our manuscript. We believe that the quality of our manuscript has been significantly improved. The following are our point-to-point responses to the reviewers' comments.

Reviewer #1 (Remarks to the Author):

Review NCOMMS-21-21769

The authors report two crystal structures of HARP dimers, one in complex with a pre-tRNA, which differ in the conformation of the domain that recognizes the tRNA elbow region. The structures provide insight into the architecture of HARP metallonucleases and tRNA recognition at the atomic level. Furthermore, a convincing model of cooperative substrate recognition by two neighboring dimers is proposed. The aforementioned aspects represent the strength of the manuscript. Several weaknesses are associated with the peripheral biochemical experiments (pre-tRNA processing experiments, size exclusion chromatography) as outlined below.

Response: We sincerely thank the reviewer for all the nice comments and criticisms. Based on the reviewer's comments and suggestions, we have carefully revised our manuscript.

Major comments

1) line 84 ("Knockout of the HARP gene ..."), rewrite e.g.: "Knockout of the HARP gene showed no growth defects in *H. volcanii* and *M. mazei*, indicating that the RNA-based RNase P is the main and essential RNase P activity in these archaea 32."

Response: Done as suggested.

Do not cite ref. 31 here; rather cite ref. 31 together with ref. 29 at the end of line 72.

Response: Done as suggested.

2) Lines 138-141: the architectural features described here are not clearly recognizable in Fig. 1 or Fig. S4B. Please harmonize text and illustrations.

Response: We sincerely thank the reviewer for the thoughtful comments. As suggested by the reviewer, we added all the secondary structural elements of HARP protein in Fig. S4B in the revised manuscript.

Line 144: the sentence " $\alpha 7$ is linked to the N-terminus of $\alpha 8$." Better say " $\alpha 7$ is separated from $\alpha 8$ by a short linker."

Response: Done as suggested.

3) Indicate $\alpha 7$ in Fig. 1G.

Response: Done as suggested. The original Fig. 1G has been moved to Fig. 4 (panel A) in the revised manuscript.

4) Line 176, change to: "... G-10 to U-2, U32 to A36 and C73/A74 located at the 5'-end, ..."

Response: Done as suggested.

5) In Fig. S3B it is not clear if pre-tRNA^{His} is cleaved specifically at the canonical RNase P cleavage site or if this is non-specific degradation. Also, at the unusually high pH of 9 of the processing buffer, metal ion-induced degradation cannot be completely excluded. These aspects represent a general weakness of the presented 10% PAA gels that do not allow resolution/identification of specific cleavage products; appropriate would be 20-25% PAA gels to resolve the 5'-leader products, using, for example, E. coli RNase P or Aq_880 processing samples as reference.

Response: We sincerely thank the reviewer for the wonderful suggestions. To verify whether pre-tRNA is cleaved by PbHARP at the correct position, we performed in vitro cleavage assays and analyzed the reaction mixtures by 20% denaturing UREA-Gels (please see Fig. S3C in the revised manuscript and the figure below). No detectable pre-tRNA cleavage could be found in the presence of cations but absence of RNase P. Aq_880 and E. coli RNase P were used as positive control in the assay. Although it was weaker than Aq_880 and E. coli RNase P, the identical sizes of the mature tRNA and 5'-leader products indicated that PbHARP cleaves pre-tRNA at the canonical cleavage site.

6) Lines 263/264: are stable hydrophobic interactions between the V107 residues evident in both, the apo enzyme and the HARP-pre-tRNA complex structure?

Response: Although they are different, the hydrophobic interactions between the V107 residues formed in both the apo-form and the tRNA-complexed PbHARP structures (please see Fig. 4A-B in the revised manuscript).

7) Paragraph beginning in line 257: here the authors compare the interface between monomers in the dimer for the apo enzyme and the dimer-tRNA complex. The reader is here forced to switch back and forth between Fig. 1G and 4A. For easier reading, move Fig. 1G to Fig. 4 and only discuss the apo form interface together with the altered interface in the dimer-tRNA complex in the context of Fig. 4.

Response: We sincerely thank the reviewer for the helpful comments. As suggested by the reviewer, we moved the original Fig. 1G to Fig. 4 (panel A) in the revised manuscript.

8) Line 280: the A118/121E mutant should be renamed to A118E/A121E. I find it surprising that this double mutant still dimerizes, as one expects disruption of the hydrophobic interface as well as electrostatic repulsion and steric clashes. Are you sure that the elution peak at 14.4 mL really represents dimers?

Response: We sincerely thank the reviewer for correcting our mistakes. "A118/121E mutant" has been replaced with "A118E/A121E mutant" in the revised manuscript. We also thank the reviewer for raising the question about the oligomerization state of the A118E/A121E mutant. As we

showed in Fig. 4C and the figure below (panel A), A118 and A121 were involved in PbHARP dimerization in the tRNA-complexed structure. However, in the apo-form structure (please see panel B of the figure below), A118 and A121 of the two HARP dimer are far away from each other and they don't participate in the dimerization of the structure. We believed that substitution of A118 and A121 with Glu residue has no impacts on PbHARP dimerization, but it will prevent the non-catalytic apo-form to the catalytic tRNA-complexed form transition of PbHARP dimer. To further confirm the oligomerization state of the A118E/A121E mutant, we performed size-exclusion chromatography for the standard marker proteins on the same column (please see Fig. S2C in the revised manuscript and the bottom panel of the figure below), comparison of the profiles suggested that the peak eluted at 14.4-mL corresponds to dimer of A118E/A121E mutant (molecular weight 47.9 KDa).

9) In general, the authors present size-exclusion chromatography profiles and claim that they represent monomers or certain oligomers (Fig. 4B, 6A, S2), but no marker proteins were run on the same column. Also, the methodological description of the size-exclusion chromatography (starting in line 554) should be completed, such that one can replicate the experiment.

Response: As suggested by the reviewer, we performed size-exclusion chromatography for the standard marker proteins on the same column (please see Fig. S2C in the revised manuscript and the bottom panel of

the figure below). The oligomerization states of the HARP proteins (WT or mutants) can be supported by their comparison with the standard proteins. The details of size-exclusion chromatography studies have been included in the revised manuscript.

10) Fig. 4C, 7D: Analysis of cleavage activity must include a positive control as part of the same experiment, to exclude that something else than the tested protein was responsible for inactivity.

Response: We have redone the cleavage reactions using wild-type PbHARP as positive control (please see Figs. 4E and 7D in the revised manuscript). The Pre-tRNA cleavage activity was observed for the wild-type PbHARP, but it was not observed for the mutants, confirming that the mutants are inactive.

11) Lines 295-303: this description of the pre-tRNA^{His} mutants is long-winded; I propose to shorten the text here by stating that variants with 7, 6 and 9 bp in the acceptor stem were constructed and to simply refer to Fig. 5A.

Response: Thanks for the great suggestions. We have shortened the paragraph and renamed the variants of pre-tRNA^{His} as His-7bp, His-6bp and His-9bp, respectively.

12) Lines 423-427, starting with "Sequence and ...": it is not immediately clear what the authors mean here by "arranged in a reversed order". It would be helpful to explain to the reader in more detail in the legend to Fig. S11 D and E what the idea is; also, mark the overlapping Arg residues (R212, R138) in Fig. S11E.

Response: "Arranged in a reversed order" mainly means that the first and second helices ($\alpha 7$ and $\alpha 8a$) of PbHARP HB domain share sequence similarity with the second and first helices (α -B and α -A) of AtPRORP PPR motifs, respectively. More details have been included in the legend to Fig. S11D, which has been renumbered as Fig. S13D in the revised manuscript.

Fig. S11E has been renumbered as Fig. S13E in the revised

manuscript. The figure only shows the structural comparison of PbHARP HB domain and AtPRORP PPR1 motif, which are most similar in sequence. Since R212 of AtPRORP belongs to the PPR4 motif of AtPRORP, it was not shown in the figure.

13) Line 456: with regard to structure and catalytic mechanism, ref. 47 is not appropriate here; I propose to cite here instead good reviews by Klemm et al. 2016, Biomolecules, doi: 10.3390/biom6020027, and Schencking et al. 2020 (https://doi.org/10.1007/978-3-030-57246-4_11), plus refs 44, 46 and Wan et al. 2019 (Cryo-electron microscopy structure of an archaeal ribonuclease P holoenzyme. Nat Commun, doi: 10.1038/s41467-019-10496-3).

Response: We sincerely thank the reviewer for the helpful suggestions and all the wonderful references, which have been cited in the revised manuscript.

14) Line 458: exchange current ref. 26 with "Holzmann et al. 2008, Cell (doi: 10.1016/j.cell.2008.09.013)"

Response: Done as suggested.

15) Line 467/468: omit "... our study further supported the evolutionary transition theory." This study in no way addresses any aspects of RNase P evolution.

Response: We thank the reviewer for the thoughtful comments. The statement has been removed in the revised manuscript.

16) Legends to figures: some figure legends are extremely parsimonious, such as those in Fig. S2 and S6. At least a remark such as "For experimental details, see Methods section of the main text" would be appropriate.

Response: We thank the reviewer for the criticisms and suggestions. Fig. S6 has been renumbered as Fig. S7 in the revised manuscript. In addition to the statement "For experimental details, see Methods section of the main text", we also included more details to the legends to the related figures.

Minor comments

- line 65: for T. brucei PRORP, cite Taschner et al. Cell Rep. 2012 (doi: 10.1016/j.celrep.2012.05.021)

Response: Thanks for the great reference, which has been cited in the manuscript.

- line 91: cite ref. 29 after "... with glucose in the medium."

Response: Done as suggested.

- line 197, rewrite to e.g. "... we showed that PbHARP is also able to process pre-tRNA^{Cys} and pre-tRNA^{Phe} under identical reaction conditions (Fig. S3B)."

Response: Done as suggested.

- line 199: "... is functionally relevant."

Response: Thanks for pointing out our mistake, which has been fixed in the manuscript.

- line 209: "... cleavage activity at all was observed for the R138A mutant."

Response: Done as suggested.

- line 223: "... D173A mutants (Fig. S6)."

Response: Thanks for the suggestion. Fig. S6 has been renumbered as Fig. S7 and cited at the end of sentence in the revised manuscript.

- line 269: replace "incorporative" with "cooperative"

Response: Thanks for the correction. The word "incorporative" has been replaced with "cooperative" in the revised manuscript.

- line 279: "...15.6 mL (Fig. 4B)."

Response: Done as suggested.

- line 294: pairing of G-1 and C72 is not visible in Fig. S5B.

Response: Done as suggested.

- lines 311-313: shorten and combine the last two sentences: "Similar to PbHARP, TrHARP failed to cleave pre-tRNA^{His-3} and also showed a clear preference for pre-tRNA^{His-1}." I also propose to rename the pre-tRNA^{His} variants to 'His 8bp', 'His 7bp' etc. for the sake of simplicity.

Response: Done as suggested.

- line 332: change Fig. S7B to Fig. S8B

Response: Thanks for pointing out our mistake, which has been fixed in the manuscript.

- lines 382-384 (last sentence before the discussion), rewrite: "Bands tentatively assigned to the PbHARP tetramer (complex of two PbHARP dimers) were clearly enhanced in the presence of pre-tRNA^{His-1}."

Response: Done as suggested.

line 387, change to: " Here we present results of structural and biochemical analyses of HARPs from four organisms, P. bacterium, T. celer. T. minervae and

T. ruber."

Response: Done as suggested.

- line 415: "... an N-terminal PPR domain."

Response: Done as suggested.

- line 466, rephrase: "... pre-tRNA-complexed structure of a full-length protein-only RNase P

Response: Done as suggested.

Reviewer #2 (Remarks to the Author):

The authors determined the first crystal structures of prokaryotic HARP enzymes with/without tRNA molecules. The structure revealed that HARP enzymes exist as a dimer mediated by two helical regions (HB region). The tRNA-complexed structure and mutational analysis showed key basic residues of HB region are involved in tRNA recognition, particularly elbow recognition. The structures also revealed that HARP undergoes conformational changes in elbow recognition and active site formation. Furthermore, the authors demonstrated HARP assemble to the oligomer (hexamer of dimer: dodecamer) by electron microscopy analysis and suggested that the oligomerization is necessary to process pre-tRNA substrate. This study provides new insights into the field of RNase P research and unveils the structural basis of how the minimal protein-only RNase P recognizes its substrates. However, several discussion points and concerns need to be addressed in a revision of the manuscript.

Response: We sincerely thank the reviewer for all the nice comments and criticisms. Based on the reviewer's comments and suggestions, we have carefully revised our manuscript.

Major issues

1. Title could be modified to convey the crystal structure of prokaryotic minimal protein-only RNase P. This article is not a review paper, but the current title is associated with all types of protein-only RNase P.

Response: Thanks for the helpful comments. The title has been replaced with "Crystal structures and insights into precursor tRNA 5'-end processing by prokaryotic minimal protein-only RNase P" in the revised manuscript.

2. The authors need a more in-depth discussion of the oligomeric state of HARP.
i) In which oligomeric state (tetramer, hexamer, or dodecamer?) do the authors think that HARP functions in vivo? The minimal catalytic unit seems to be a tetramer, but TrHARP assembles to the dodecamer.

Response: We agree with the reviewer that TrHARP can assemble into dodecamer. The recently reported Aq_880 and Halorhodospira halophila SL1 HARP also assemble as dodecamer. However, as demonstrated by size-exclusion chromatography, PbHARP exists as a dimer and the oligomerization state of TcHARP is changable. We believed that tetramer is the minimal catalytic unit of HARPs in vivo, whereas dodecamerization may allow HARP to bind and cut multiple pre-tRNA simultaneously. We have reflected these opinions in the revised manuscript.

ii) What is the significance of dodecamer?

Response: As molded by Teramoto and coworkers in recent reference

{Teramoto, T. et al. Minimal protein-only RNase P structure reveals insights into tRNA precursor recognition and catalysis. J Biol Chem 297, 101028, doi:10.1016/j.jbc.2021.101028 (2021) }, HARP dodecamer could bind up to 10 pre-tRNAs. Compared to tetramer, HARP dodecamer might be more efficient in pre-tRNA binding and cleavage.

iii) How many molecules of pre-tRNA can bind to the HARP dodecamer? The big picture would need to be discussed.

Response: As molded by Teramoto and coworkers in recent reference {Teramoto, T. et al. Minimal protein-only RNase P structure reveals insights into tRNA precursor recognition and catalysis. J Biol Chem 297, 101028, doi:10.1016/j.jbc.2021.101028 (2021) }, HARP dodecamer could bind up to 10 pre-tRNAs. Since this model has been published, we did not repeat it in our manuscript.

iv) Is the low activity of PbHARP related to its difficulty in oligomerization? What is the difference in the structural basis between high and low activity HARPs? Difference of residues?

Response: As we observed in the Pre-tRNA complexed structure, PbHARP can form oligomer. However, different from other HARPs, PbHARP requires high concentration (>300 mM) of salt in buffer to be stable. High concentration of salt may cause the dissociation of PbHARP oligomer and lead to the apparent dimer in solution. Like other HARPs, the pre-tRNA cleavage activity of PbHARP was measured in buffer containing 100 mM NaCl. Under such condition, PbHARP is less stable than other HARPs, which might be the main reason of the low activity of PbHARP.

v) In crystal packing, were there any interactions similar to the oligomeric state?

Response: We sincerely thank the reviewer for raising this question. Via structural superposition, we found that the packing interactions between the AB and CD dimers in the pre-tRNA complexed PbHARP structure are very similar to these formed by Aq_880 dimers in the cryo-EM structure (please see the figure below for the comparison). The interactions are mainly mediated by the residues 80-87 and 185-189 in PbHARP structure. Sequence alignment (please see Fig. S1) suggested that these residues are conserved in HARPs.

vi) Very recently, other HARP dodecamer papers have been published, so please cite them for a better discussion.

Response: *Thanks for reminding of the published HARP structures. The related references {Teramoto, T. et al. Minimal protein-only RNase P structure reveals insights into tRNA precursor recognition and catalysis. J Biol Chem 297, 101028, doi:10.1016/j.jbc.2021.101028 (2021); Feyh, R. et al. Structure and mechanistic features of the prokaryotic minimal RNase P. Elife 10, doi:10.7554/eLife.70160 (2021)} have been cited and discussed in the manuscript.*

3. Since the 5' leader sequence is not in the active site, the authors performed mutational analyses to eliminate the effect of crystal packing and to evaluate the important basic residues. However, the binding constants should be obtained for some representative residues instead of qualitative cleavage assays. With qualitative cleavage assays alone, readers cannot rigorously assess the importance of residues.

Response: *We sincerely thank the reviewer for the helpful comments. To determine the pre-tRNA binding constants of WT and mutated PbHARP proteins, we tried both ITC and MST assays. Unfortunately, we failed to obtain any reliable results. In the assay buffer containing low concentration (100 mM) of salt, the proteins become unstable. The proteins are stable in the buffer containing high concentration (>300 mM) of salt, but the salt interferes with PbHARP oligomerization and substrate binding.*

To investigate the functional importance of the key pre-tRNA recognizing residues, we then performed in vitro EMSA assay (please see Fig. S6A in the revised manuscript and panel A of the figure below). We found that Ala substitution of the pre-tRNA recognizing residues has no

strong impacts on pre-tRNA binding by PbHARP. In combination with the cleavage assay results, we believed that these residues are critical for the assembly of the catalytic form complex.

In addition, we also performed *in vitro* EMSA assays for the L103E/V107E and A118E/A121E mutants (please see Fig. S6B in the revised manuscript and panel B of the figure below). Compared to WT PbHARP, the pre-tRNA binding abilities of the two mutants, especially L103E/V107E, are weaker. In combination with the structures, size-exclusion chromatography and cleavage assays, these results suggested that L103E/V107E and A118E/A121E have difficulty in forming catalytic dimer.

We are truly sorry that we could not get more accurate values for the binding constants of WT and mutated PbHARP proteins. However, we believed that our *in vitro* EMSA binding and cleavage assay results could explain the functional importance of the key residues to some extent.

4. Since the structures of PbHARP and TcHARP are the first crystal structures of HARP family, the authors would need to describe the detail of the active site.
 i) The authors described only three residues (D151, D155, and D173) of the active site. Can authors propose other key residues from the structural information? Are there any other residues in the active site?

Response: We thanks the reviewer for the helpful comments. In addition to the three catalytic residues D151, D155 and D173, two Arg residues (R108 and R146) near the active site are also critical for the function of PbHARP. Their conformation and impacts on the Pre-tRNA cleavage activity of PbHARP have been demonstrated in the Figs. 7C-D in the manuscript.

ii) The author described the Mg ion essential for catalysis is coordinated with D151, D155, and D173, but NO rationale or previous research is provided in the manuscript.

Response: The functional importance of the three catalytic Asp residues can be supported by the mutagenesis and in vitro cleavage assays in our study and previous study {Nickel, A. I. et al. Minimal and RNA-free RNase P in Aquifex aeolicus. P Natl Acad Sci USA 114, 11121-11126, doi:10.1073/pnas.1707862114 (2017)}. The functional importance of Mg can be supported by adding EDTA {Teramoto, T. et al. Minimal protein-only RNase P structure reveals insights into tRNA precursor recognition and catalysis. J Biol Chem 297, 101028, doi:10.1016/j.jbc.2021.101028 (2021)} or lacking of Mg in the reaction condition (this study). Mg coordination with the catalytic Asp residues is mainly supported by the cation ions captured in the active site of PbHARP in the Pre-tRNA complexed structure and the structures of other cation-dependent nucleases (such as RNase III, RNase D and RNase H), which have been cited in the manuscript. Of course, one catalytic form HARP-pretRNA-Mg ternary complex will provide more direct evidence for the coordination. However, we do not have such complex right now.

iii) In the structure of tRNA complex, Ca ions are bound to the active site. What is the binding position of the Ca ion compared to the Ca, Mg, and Mn ions in the previously determined crystal structure of Arabidopsis thaliana PRORPs?

Response: In the reported AtPRORP1 structure (PDB_ID: 4g24), Mn ions only coordinate with the side chains of the catalytic Asp residues. However, in addition to the side chain of D151, Ca ion also coordinates with the main chain O atoms of R146 and I149 in the pre-tRNA complexed PbHARP structure (please see Fig. 3B). Due to the lack of catalytic HARP-pretRNA-Cation ternary complex, coordination between Ca (or Mg) with the other two catalytic Asp residues (D155 and D173) is not clear right now.

We sincerely thank the reviewer for the kind understanding that we did not compare the cation coordination of HARP and AtPRORP in this manuscript. We are trying to capture the catalytic HARP-pretRNA-Cation complex using HARP from other species. Hopefully, we can solve the structure and unravel the detailed coordination soon.

5. The authors confirmed the oligomeric state of HARPs by size-exclusion chromatography. However, they have not shown the standard protein. Thus, the readers do not know how the peak positions of the dimer, oligomer, or monomer were determined.

i) The authors would need to illustrate standard peak position and calibration curve.

Response: Thanks for the helpful suggestions. We have performed size-exclusion chromatography for the standard marker proteins on the same column (please see Fig. S2C and the figure below). The oligomerization states of the HARP proteins can be supported by their comparison with the standard proteins.

ii) Peak profile of L103E/V107E indicated a broad peak and low absorbance. The authors would need to check protein stability. The activity may be lost due to low stability.

Response: We sincerely thank the reviewer for the helpful suggestion. Following the reviewer's suggestion, we measured the melting temperatures of WT PbHARP and the L103E/V107E mutant. The results showed that the L103E/V107E mutant is less stable (please see Fig. S9 in the revised manuscript and the figure below). We agree with the reviewer that the low stability might be one reason for the lack of pre-tRNA cleavage activity of the L103E/V107E mutant. Like the A118E/A121E mutant, we believe that failure to form functional dimer also impairs the pre-tRNA cleavage activity of the L103E/V107E mutant. We have included these opinions in the revised manuscript.

Minor points

Ln24-25: Knowledge of substrate recognition and cleavage mechanism of eukaryotic PRORPs has been accumulated. It would be better to limit to HARP.

Response: Thanks for the suggestion. We has replaced the sentence with “To date, various PRORPs have been discovered, but the basis underlying substrate binding and cleavage by HARPs (homolog of Aquifex RNase P) remains elusive.” in the revised manuscript.

Ln59-67: It is a little difficult to understand the relationship between MRPP3 and PRORP.

Response: In human mitochondrion, pre-tRNA 5'-end is cleaved by the PRORP complex, which is composed of MRPP1, MRPP2 and MRPP3. MRPP3 is the catalytic subunit of human mitochondrial PRORP complex and is homologous to PRORP from Arabidopsis thaliana (A. thaliana) and Trypanosoma brucei (T. brucei).

Ln68: MRPP3-like PRORP brings up the image of PRORP that function with three subunits. Readers may mistakenly think that bacteria have Arabidopsis and Trypanosoma-type PRORP.

Response: We sincerely thank the reviewer for the helpful comments. To avoid the possible confusion, we have replaced the sentence with “Bacteria and Archaea do not express any AtPRORP, TbPRORP or human MRPP3-like protein” in the revised manuscript.

Ln83-84: The overall preservation of HARP does not appear to be very limited.
Response: We agree with the reviewer that the overall preservation of HARP is not very limited. We have replaced the sentence with “Aq_880 shares some conserved residues with HvHARP, MmHARP and HARPs from many other species (Fig. S1A).” in the revised manuscript.

Ln90: lack of reference

Response: One reference {Nickel, A. I. et al. Minimal and RNA-free RNase P in Aquifex aeolicus. P Natl Acad Sci USA 114, 11121-11126, doi:10.1073/pnas.1707862114 (2017)} has been cited at the end of the sentence.

Ln145: Please show b-factor distribution or weak electron density in supplemental figure

Response: We sincerely thank the reviewer for the thoughtful comments. One new figure has been included in Fig. S4 (panel C) in the revised manuscript to show the flexibility of the $\alpha 7$ - $\alpha 8$ connecting linker.

Ln161-162: Please describe the r.m.s.d.

Response: Thanks for the suggestion. The RMSD values between TcHARP and PbHARP monomers have been provided in the revised manuscript.

Ln244: Could the tRNA-complex be described as the active state even though it is not active because Ca is bound to the active site?

Response: Different from PbHARP in the apo-form structure, PbHARP in the tRNA-complexed structure is suitable for substrate and cation binding. Upon the substitution of Ca with Mg, the protein will catalyze pre-tRNA cleavage reaction. Therefore, we described the structure as the active state, mainly emphasizing the conformation of the protein.

Ln288-313: It might be difficult to follow the relationship between the name of the pre-tRNA^{His} variant and the number of Base pairs. Please give them names that are easy to understand.

Response: Thanks for the helpful comments. As suggested by the reviewer #1, we have renamed the three pre-tRNA^{His} variants as His-7bp, His-6bp and His-9bp to indicate that they have seven, six and nine base pairs at the acceptor stem regions, respectively.

Ln320: lack of reference

Response: The reference {Nickel, A. I. et al. Minimal and RNA-free RNase P in Aquifex aeolicus. P Natl Acad Sci USA 114, 11121-11126, doi:10.1073/pnas.1707862114 (2017)} has been cited at the end of the

sentence.

Ln320: Why didn't TcHARP check the oligomeric state by EM? A reason would be welcome.

Response: *We thank the reviewer for raising this interesting question. Different from Aq_880 and TrHARP, the oligomerization state of TcHARP is changeable (please see Fig. S2B), therefore we did not check its oligomerization state by EM previously. Following the reviewer's suggestion, we performed EM study for TcHARP (please see the figure below, panel A). Different from Aq_880 and TrHARP, TcHARP did not form ring-shaped dodecamer. Instead, they formed some half-circled particles (please see the panel B of the figure below), which are likely composed of three or four TcHARP dimer. Unfortunately, we could not obtain enough well-organized particles to build a convincing model for TcHARP at this stage. Therefore, we did not include these results in the revised manuscript.*

Ln380: The authors should provide a brief description of the cross-linking details. Readers don't know what is being cross-linked.

Response: *The crosslinking reagent BS3 can crosslink the side chains of two lysine residues together. We have included these information in the revised manuscript.*

Ln465-466: The authors described that the PbHARP/pre-tRNA^{His} structure represents the only pre-tRNA-complexed structure of full-length PRORP. However, the 5' leader sequence is not in the active site. it would be better to rephrase it.

Response: *Thanks for the reminder. We have replaced the sentence with "the PbHARP/pre-tRNA^{His} structure represents the only tRNA-complexed structure of a full-length protein-only RNase P" in the revised manuscript.*

Figs: Please show the cleavage percentage at the bottom of the gel figures.

Response: Done as suggested.

Fig.1: MRPP1/2 proteins might not be necessary in the main figure.

Response: MRPP1/2 proteins have been removed from the main figure in the revised manuscript.

Fig.S7: Please illustrate Fo-Fc omit map of Ca ion. Model bias might be a concern.

Response: Thanks for the suggestion. The Fo-Fc omit map of Ca ion has been included in the Fig. S7, which has been renumbered as Fig. S8 in the revised manuscript.

Fig.S9: Please show sizes of protein markers.

Response: The sizes of protein markers have been included in the revised manuscript.

Fig.Ss: The extra lines in the figure outline are seen.

Response: We thank the reviewer for the kindly reminder. We have updated the figures to eliminate the outline.

REVIEWER COMMENTS

Reviewer #1 (Remarks to the Author):

The authors have addressed all experimental issues satisfactorily, which largely improved the quality of the manuscript. After careful rereading, several minor issues remain where (A) statements should be set right or rewritten for better understandability, in addition to (B) several 'language polishing' suggestions. I also proposed changes/reference additions at two locations in the Discussion to better appreciate the two recently reported cryo-EM structures of HARP dodecamers. Overall, there is little comparative discussion of the findings of the two HARP cryo-EM structures, which is still a weakness of the manuscript.

A. Rephrasing of statements:

- Line 69, rewrite: "Bacteria and Archaea do not express eukaryotic-like PRORP proteins, but ..."
- Line 73: "... are also present in some other bacteria and are even more wide-spread in Archaea^{30,31}."
- Line 84: add ref. 33 after 'euryarchaeotes'
- Line 85: delete 'some'
- Lines 93-95: delete sentence "However, likely due to their important role in tRNA maturation, no HARP gene knockout strains has been reported for any Aquificaceae." (knockouts have not been tested in members of the Aquificaceae because these bacteria are not genetically manipulable)
- line 102: "... into oligomers. Structural analysis, mutagenesis, in vitro cleavage and crosslinking assay all suggested a HARP pre-tRNA binding and cleavage mode involving the cooperation of two dimers."
- Line 114, rewrite to: "... similarities with Aq_880, and the sequence identity is about 30% among the five proteins (Fig. S1B)." (30% identity is not necessarily very low among orthologous proteins)
- Line 181, rephrase: "The asymmetric dimerization orients the side chains of four HB domain Arg residues, including R116 and R123 of monomer B and R138 and R142 of monomer A, toward the pre-tRNA His T Ψ C-loop where they form extensive H-bond interactions with the phosphate backbones of G52-G56 (Fig. 2B)."
- Line 204, replace sentence beginning with 'Although': "Two 5'-leader cleavage products were observed, one being identical in size to that generated by Aq_880 and E. coli RNase P (Fig. S3C, outer right lane). This shows that PbHARP cleaves pre-tRNA at the canonical cleavage site in addition to some apparent miscleavage at the neighboring upstream site."
- Line 219, rewrite: "... are critical for assembly of the catalytically active PbHARP/pre-tRNA complex."
- Line 226, rephrase to: "The functional importance of these catalytic residues could be confirmed by ..." (their catalytic importance was already shown in ref. 30)
- Line 251: "Although not observed in the structure, this arrangement may allow PbHARP to coordinate a second divalent cation, thereby utilizing the common two metal ion-assisted mechanism of catalysis^{35,37,38}."
- Line 254: "Divalent metal ion coordination, ..."
- Line 282: Here the authors state "However, the C-termini of α 7 helices gathered together in the

complex structure, ...". This is not immediately evident from Fig. 2. I propose to refer here also to Fig. S5 that illustrates this well.

- Line 454, rewrite: "The total size of HARP is only about 1/3 of AtPRORP1 and HsMRPP3, the latter even requiring two additional protein cofactors (HsMRPP1 and HsMRPP2) for in vivo activity. However, HARP performs the same reaction as AtPRORP1 and the HsMRPP complex. A similar phenomenon has also observed ..."

- Line 465: "Our and the related recent studies^{52,53} of HARP support the idea that oligomerization and division of function between mono- or dimers may be a more common way (than previously thought) to provide small enzymes with substrate binding and catalytic abilities."

- Line 473: the ref. numbers must be 28,29,51

- Line 487: We believe that a tetramer is the minimal catalytic unit of HARP in vivo, whereas dodecamerization may allow HARP to bind and cut multiple pre-tRNAs simultaneously, as proposed recently^{52,53}.

(add here also ref. 53!)

- Line 489: "Our study reveals the molecular basis for substrate binding and cleavage by HARP proteins."

- Lines 495, 501,525, 538,542, 546, 579, Table S2: the genes for RNA-based bacterial RNase P are written as *rnpA* and *rnpB* (in italics); the RNA itself is named 'RNase P RNA' or 'P RNA'; the RNase P protein should be written as 'RnpA'. Please change correspondingly.

- Line 501: "... transcription of E. coli RNase P RNA, the *rnpB* gene with T7 promoter and hammerhead ribozyme (Table S2) was inserted into a pUC18 vector at the HindIII and EcoRI restriction sites."

- Lines 585-587: the addition and composition of gel loading buffer is missing in the method description. Legend to Fig. S7, line 70, rewrite: "... of divalent metal ions (Mg²⁺) and the catalytic Asp residues of PbHARP."

B. Linguistic/grammatical corrections:

- Line 99: "... showed that HARP is able to undergo large conformational changes that facilitate pre-tRNA binding and catalytic site formation."

- Line 120: "... it exists as dimer; however, it assembles into larger oligomers in buffer containing 100 mM salt."

- Line 149: "... forming a four-helix bundle ..."

- Line 153: "β4 plays an important ..."

- Line 207: "... are functionally ..."

- Line 231: "... about 60% of the substrate were cleaved; only trace amounts of intact substrate remained ..."

- Line 238: "... In contrast, the side chain of D151 points away from D155, the averaged distance between their carboxylate groups exceeding 7 Å."

- Line 244: "... As demonstrated for many nucleases, ..."

- Line 264: "... α8 is split into ..."

- Line 285: "In the complex structure, the A118 residues center around the HB domain central axis, the

distance between their side chains is only 3.6 Å (Fig. 4C)."

- Line 296: "... we also constructed the A118E/A121E mutant ..."

- Line 300: "... is weak; no pre-tRNA binding at all was observed ..."

- Line 302: "... but detectable pre-tRNA^{His} cleavage activity was neither observed for the L103E/V107E nor for the A118E/A121E mutant under identical reaction conditions (Fig. 4E), probably due to their low stability and/or difficulty to form a functional dimer."

- Line 308: "In the structure of the complex, PbHARP mainly recognizes the shape and backbone of the TΨC-loop (Fig. 2B-E), a structural element conserved in all tRNAs."

- Line 313: "... 5'-end maturation, the phosphodiester between nucleotides -1 and +1 is usually cleaved."

- Line 321: "... is much weaker than those of His-7bp and His-6bp. In fact, the cleavage efficiency of His-7bp was comparable to those of ..."

- Line 329: "Both TrHARP and PbHARP preferably cleaved the His-7bp variant (Fig. 5B-C), but their oligomerization states are different. TrHARP eluted from the size exclusion column at ~11.4 mL in all buffers containing 100, 200 or 500 mM salt (Fig. 6A). A previous study suggested the possibility that Aq_880 may form either six trimers or three hexamers³⁰ (inferred from size exclusion chromatography profiles), but both PbHARP and TcHARP formed dimers in the crystal structures (Fig. 1C-F)."

- Line 335: "... was subjected ..."

- Line 340: "... shape and size of the ring matched well with an assembly of six PbHARP dimers that arrange in a left-handed orientation (Fig. 6D)."

- Line 347: "... are ~50° between Dimer_1 and Dimer_2, Dimer_2 and Dimer_3, and ..."

- Line 360: "... with TrHARP suggested that PbHARP has the potential to form larger oligomers under proper conditions. The incorporation of additional PbHARP dimers can be easily modeled by rotating ..."

- Line 372: "G-1 pairs with C72 in native pre-tRNA^{His}, and such pairing may interfere with the conformational change and metal ion coordination at the scissile phosphodiester between G-1 and G1. The acceptor stem of His-9bp is 2 bp longer ..."

- Line 386: "... had no strong impact on ..."

- Line 389: "Pre-tRNA^{His} is recognized by the PbHARP AB dimer in the complex, but neither R108 nor R146 of the AB dimer are involved in the interaction. These structural observations combined with the inactivity of these mutants further suggested that PbHARP binds and cleaves pre-tRNA in a cooperative mode. To better demonstrate the cooperation between PbHARP dimers, we performed ..."

- Line 394: "In addition to the monomer, ..."

- Line 397: "... of two lysine residues. Although not clearly detectable in the absence of His-7bp, ..."

- Line 407: "However, unlike PRORP enzymes constituting the PRORP subfamily, HARPs belong to the PIN_5 cluster subgroup."

- Line 415: "The activities of HARP ..."

- Line 416: "The apo-form HsMRPP3 structure is in an inactive form, ..."

- Line 419: "... and TcHARP are also in an inactive form."

- Line 421: "... suggested that HARP undergo self-activation via a large conformational change to the catalytically active form (Fig. 3B-D)."

- Line 426: "However, the third catalytic Asp residue (D151) of PbHARP is positioned in mirror image to D399 of AtPRORP1 (Fig. S12E). This mirror-image position of D151/D399 and the self-activation ability distinguish HARP from eukaryotic PRORPs."

- Line 436: "... HARP and AtPRORP1 PPR both recognize ..."
- Line 439: "... helices of the HARP HB and the AtPRORP1 PPR domain ..."
- Line 445: "... central domain at the bottom. Although it has not been experimentally verified, it is likely that the V-shaped conformation ..."
- Line 449: "Neither a single monomer nor dimer of HARP ..."
- line 451: "... results suggest that ..."
- Line 452: "Interestingly, although not identical to AtPRORP1, the two HARP dimers also seem to adopt a V-shaped conformation (Fig. 7B)."
- Line 459: "... functions as a homodimer."
- Line 474: "... archaea and bacteria that also encode ..."
- Line 480: "... assembled into dodecameric structures. The PbHARP and TcHARP structures we report here are the only available crystal structures of HARP."
- Line 484: "Instead of being dodecamers, PbHARP and TcHARP exist ..."
- Line 489: "... pre-tRNAs simultaneously ..."
- Line 537: "The template that contains a T7 promoter sequence upstream of the pre-tRNA or RNase P RNA coding region was obtained by overlap PCR using the primers listed in Table S4."
- Line 541: "To remove the hammerhead ribozyme and increase RNase P RNA yield, the DNase I-digested transcription sample was incubated overnight at 4°C."
- Line 558: "Assays with a final volume of 10 μ L were set up. The reaction mixtures that included 1 μ M pre-tRNA and protein serially diluted from 1 μ M to 10 μ M were incubated ..."
- Line 563: "... 4°C) in 1 \times TAE buffer."
- Line 580: delete 'including'
- Line 584: "... were 0.25 μ M, 5 μ M and 5 mM, ..."
- Line 585: "The reactions were terminated after 15 min by heating to 95°C for 5 min."
- Line 741: replace C) with E)

Reviewer #2 (Remarks to the Author):

The authors have answered most of my questions and modified their text accordingly. I feel that the authors are honestly and carefully explaining and interpreting the new data (e.g. EMSA, nano-DSF) in the response letter. However, these explanations and interpretations are not reflected and are inadequate in the revised manuscript. Those should be clearly addressed in the manuscript.

The following two points are particularly important.

1. Interpretation or explanation of discrepancies between EMSA data and cleavage assay data for the mutants
2. Monomerization by the L103E/V107E mutation affects protein stability? or the L103E/V107E mutation itself affect protein stability?

Minor point

- i) The domain architectures of AtPRORP1 in Fig1 A lacks the Nuclease domain.
- ii) In the figure legend of Fig 4, there are two Cs and no E.

We sincerely thank the reviewers for rereading our manuscript with great care, we would also like to thank the reviewers for their very helpful comments, encouragements and criticisms as well. On the basis of their comments and suggestions, we have carefully revised our manuscript and addressed all the remaining issues raised by the reviewers. We believe that the quality of our manuscript has been significantly improved. The following are our point-to-point responses to the reviewers' comments.

Reviewer #1 (Remarks to the Author):

The authors have addressed all experimental issues satisfactorily, which largely improved the quality of the manuscript. After careful rereading, several minor issues remain where (A) statements should be set right or rewritten for better understandability, in addition to (B) several 'language polishing' suggestions. I also proposed changes/reference additions at two locations in the Discussion to better appreciate the two recently reported cryo-EM structures of HARP dodecamers. Overall, there is little comparative discussion of the findings of the two HARP cryo-EM structures, which is still a weakness of the manuscript.

Response: We sincerely thank the reviewer for all the nice comments and criticisms. Based on the reviewer's comments, we have included one additional figure (please see Fig. S14 and the figure below) to compare our and the reported cryo-EM structure of HARP in the revised manuscript.

A. Rephrasing of statements:

- Line 69, rewrite: "Bacteria and Archaea do not express eukaryotic-like PRORP proteins, but ..."

Response: Done as suggested.

- Line 73: "... are also present in some other bacteria and are even more widespread in Archaea^{30,31}."

Response: Done as suggested.

- Line 84: add ref. 33 after 'euryarchaeotes'

Response: Thanks for the suggestion. Ref. 33 has been cited after

'euryarchaeotes' in the revised manuscript.

- Line 85: delete 'some'

Response: Done as suggested.

- Lines 93-95: delete sentence "However, likely due to their important role in tRNA maturation, no HARP gene knockout strains has been reported for any Aquificaceae." (knockouts have not been tested in members of the Aquificaceae because these bacteria are not genetically manipulable)

Response: We thank the reviewer for the thoughtful comments. The statement has been removed in the revised manuscript.

- line 102: "... into oligomers. Structural analysis, mutagenesis, in vitro cleavage and crosslinking assay all suggested a HARP pre-tRNA binding and cleavage mode involving the cooperation of two dimers."

Response: Done as suggested.

- Line 114, rewrite to: "... similarities with Aq_880, and the sequence identity is about 30% among the five proteins (Fig. S1B)." (30% identity is not necessarily very low among orthologous proteins)

Response: We thank the reviewer for the suggestion. This sentence has been rewritten in the revised manuscript.

- Line 181, rephrase: "The asymmetric dimerization orients the side chains of four HB domain Arg residues, including R116 and R123 of monomer B and R138 and R142 of monomer A, toward the pre-tRNA^{His} T ψ C-loop where they form extensive H-bond interactions with the phosphate backbones of G52-G56 (Fig. 2B)."

Response: Done as suggested.

- Line 204, replace sentence beginning with 'Although': "Two 5'-leader cleavage products were observed, one being identical in size to that generated by Aq_880 and E. coli RNase P (Fig. S3C, outer right lane). This shows that PbHARP cleaves pre-tRNA at the canonical cleavage site in addition to some apparent miscleavage at the neighboring upstream site."

Response: Done as suggested.

- Line 219, rewrite: "... are critical for assembly of the catalytically active PbHARP/pre-tRNA complex."

Response: Done as suggested.

- Line 226, rephrase to: "The functional importance of these catalytic residues could be confirmed by ..." (their catalytic importance was already shown in ref. 30)

Response: Thanks for the thoughtful suggestion. This sentence has been rephrased in the revised manuscript.

- Line 251: “Although not observed in the structure, this arrangement may allow PbHARP to coordinate a second divalent cation, thereby utilizing the common two metal ion-assisted mechanism of catalysis^{35,37,38}.”

Response: Done as suggested.

- Line 254: “Divalent metal ion coordination, ...”

Response: Done as suggested.

- Line 282: Here the authors state “However, the C-termini of $\alpha 7$ helices gathered together in the complex structure, ...”. This is not immediately evident from Fig. 2. I propose to refer here also to Fig. S5 that illustrates this well.

Response: We sincerely thank the reviewer for the great suggestion. In addition to Fig. 2, Fig. S5A and S5C have also been cited in the revised manuscript.

- Line 454, rewrite: “The total size of HARP is only about 1/3 of AtPRORP1 and HsMRPP3, the latter even requiring two additional protein cofactors (HsMRPP1 and HsMRPP2) for in vivo activity. However, HARP performs the same reaction as AtPRORP1 and the HsMRPP complex. A similar phenomenon has also been observed ...”

Response: Done as suggested.

- Line 465: “Our and the related recent studies 52,53 of HARP support the idea that oligomerization and division of function between mono- or dimers may be a more common way (than previously thought) to provide small enzymes with substrate binding and catalytic abilities.”

Response: Thanks for the suggestion. The recently reported HARP structures were mentioned here. The references were numbered 44 and 45 in the revised manuscript.

- Line 473: the ref. numbers must be 28,29,51

Response: Thanks for the correction. The references have been updated in the revised manuscript.

- Line 487: We believe that a tetramer is the minimal catalytic unit of HARP in vivo, whereas dodecamerization may allow HARP to bind and cut multiple pre-tRNAs simultaneously, as proposed recently 52,53.

(add here also ref. 53!)

Response: Thanks for the suggestion. Reference 52 and 53 have been renumbered as 44 and 45, and both references were cited at the end of the sentence in the revised manuscript.

- Line 489: "Our study reveals the molecular basis for substrate binding and cleavage by HARP proteins."

Response: Done as suggested.

- Lines 495, 501,525, 538,542, 546, 579, Table S2: the genes for RNA-based bacterial RNase P are written as *rnpA* and *rnpB* (in italics); the RNA itself is named 'RNase P RNA' or 'P RNA'; the RNase P protein should be written as 'RnpA'. Please change correspondingly.

Response: We sincerely thank the reviewer for pointing out our mistakes. Based on the reviewer's suggestions, we have carefully examined and corrected all the mistakes in the revised manuscript.

- Line 501: "... transcription of E. coli RNase P RNA, the *rnpB* gene with T7 promoter and hammerhead ribozyme (Table S2) was inserted into a pUC18 vector at the HindIII and EcoRI restriction sites."

Response: Done as suggested.

- Lines 585-587: the addition and composition of gel loading buffer is missing in the method description.

Response: Thanks for the helpful suggestion. The addition and the detailed composition of gel loading buffer have been included in the revised manuscript.

Legend to Fig. S7, line 70, rewrite: "... of divalent metal ions (Mg^{2+}) and the catalytic Asp residues of PbHARP."

Response: Done as suggested.

B. Linguistic/grammatical corrections:

- Line 99: "... showed that HARP is able to undergo large conformational changes that facilitate pre-tRNA binding and catalytic site formation."

Response: Done as suggested.

- Line 120: "... it exists as dimer; however, it assembles into larger oligomers in buffer containing 100 mM salt."

Response: Done as suggested.

- Line 149: "... forming a four-helix bundle ..."

Response: Done as suggested.

- Line 153: " $\beta 4$ plays an important ..."

Response: Done as suggested.

- Line 207: "... are functionally ..."

Response: Done as suggested.

- Line 231: "... about 60% of the substrate were cleaved; only trace amounts of intact substrate remained ..."

Response: Done as suggested.

- Line 238: "... In contrast, the side chain of D151 points away from D155, the averaged distance between their carboxylate groups exceeding 7 Å."

Response: Done as suggested.

- Line 244: "... As demonstrated for many nucleases, ..."

Response: Done as suggested.

- Line 264: "... $\alpha 8$ is split into ..."

Response: Done as suggested.

- Line 285: "In the complex structure, the A118 residues center around the HB domain central axis, the distance between their side chains is only 3.6 Å (Fig. 4C)."

Response: Thanks for the suggestion. Fig. 4C has been cited at the end of this sentence in the revised manuscript.

- Line 296: "... we also constructed the A118E/A121E mutant ..."

Response: Done as suggested.

- Line 300: "... is weak; no pre-tRNA binding at all was observed ..."

Response: Done as suggested.

- Line 302: "... but detectable pre-tRNA^{His} cleavage activity was neither observed for the L103E/V107E nor for the A118E/A121E mutant under identical reaction conditions (Fig. 4E), probably due to their low stability and/or difficulty to form a functional dimer."

Response: Done as suggested.

- Line 308: "In the structure of the complex, PbHARP mainly recognizes the shape and backbone of the T ψ C-loop (Fig. 2B-E), a structural element conserved in all tRNAs."

Response: Done as suggested.

- Line 313: "... 5'-end maturation, the phosphodiester between nucleotides -1 and +1 is usually cleaved."

Response: Done as suggested.

- Line 321: "... is much weaker than those of His-7bp and His-6bp. In fact, the cleavage efficiency of His-7bp was comparable to those of ..."

Response: Done as suggested.

- Line 329: "Both TrHARP and PbHARP preferably cleaved the His-7bp variant (Fig. 5B-C), but their oligomerization states are different. TrHARP eluted from the size exclusion column at ~11.4 mL in all buffers containing 100, 200 or 500 mM salt (Fig. 6A). A previous study suggested the possibility that Aq_880 may form either six trimers or three hexamers³⁰ (inferred from size exclusion chromatography profiles), but both PbHARP and TcHARP formed dimers in the crystal structures (Fig. 1C-F)."

Response: Done as suggested.

- Line 335: "... was subjected ..."

Response: Done as suggested.

- Line 340: "... shape and size of the ring matched well with an assembly of six PbHARP dimers that arrange in a left-handed orientation (Fig. 6D)."

Response: Done as suggested.

- Line 347: "... are ~50° between Dimer_1 and Dimer_2, Dimer_2 and Dimer_3, and ..."

Response: Done as suggested.

- Line 360: "... with TrHARP suggested that PbHARP has the potential to form larger oligomers under proper conditions. The incorporation of additional PbHARP dimers can be easily modeled by rotating ..."

Response: Done as suggested.

- Line 372: "G-1 pairs with C72 in native pre-tRNA^{His}, and such pairing may interfere with the conformational change and metal ion coordination at the scissile phosphodiester between G-1 and G1. The acceptor stem of His-9bp is 2 bp longer ..."

Response: Done as suggested.

- Line 386: "... had no strong impact on ..."

Response: Done as suggested.

- Line 389: "Pre-tRNA^{His} is recognized by the PbHARP AB dimer in the complex, but neither R108 nor R146 of the AB dimer are involved in the interaction. These structural observations combined with the inactivity of these mutants further suggested that PbHARP binds and cleaves pre-tRNA in a cooperative mode. To better demonstrate the cooperation between PbHARP dimers, we performed ..."

Response: Done as suggested.

- Line 394: "In addition to the monomer, ..."

Response: Done as suggested.

- Line 397: "...of two lysine residues. Although not clearly detectable in the absence of His-7bp, ..."

Response: Done as suggested.

- Line 407: "However, unlike PRORP enzymes constituting the PRORP subfamily, HARPs belong to the PIN_5 cluster subgroup."

Response: Done as suggested.

- Line 415: "The activities of HARP ..."

Response: Thanks for the correction. The word "activity" has been changed into it's plural form.

- Line 416: "The apo-form HsMRPP3 structure is in an inactive form, ..."

Response: Done as suggested.

- Line 419: "... and TcHARP are also in an inactive form."

Response: Done as suggested.

- Line 421: "... suggested that HARP undergo self-activation via a large conformational change to the catalytically active form (Fig. 3B-D)."

Response: Done as suggested.

- Line 426: " However, the third catalytic Asp residue (D151) of PbHARP is positioned in mirror image to D399 of AtPRORP1 (Fig. S12E). This mirror-image position of D151/D399 and the self-activation ability distinguish HARP from eukaryotic PRORPs."

Response: Done as suggested.

- Line 436: "... HARP and AtPRORP1 PPR both recognize ..."

Response: Done as suggested.

- Line 439: "... helices of the HARP HB and the AtPRORP1 PPR domain ..."

Response: Done as suggested.

- Line 445: "... central domain at the bottom. Although it has not been experimentally verified, it is likely that the V-shaped conformation ..."

Response: Done as suggested.

- Line 449: "Neither a single monomer nor dimer of HARP ..."

Response: Done as suggested.

- line 451: "... results suggest that ..."

Response: Done as suggested.

- Line 452: "Interestingly, although not identical to AtPRORP1, the two HARP dimers also seem to adopt a V-shaped conformation (Fig. 7B)."

Response: Done as suggested.

- Line 459: "... functions as a homodimer."

Response: Done as suggested.

- Line 474: "... archaea and bacteria that also encode ..."

Response: Done as suggested.

- Line 480: "... assembled into dodecameric structures. The PbHARP and TcHARP structures we report here are the only available crystal structures of HARP."

Response: Done as suggested.

- Line 484: "Instead of being dodecamers, PbHARP and TcHARP exist ..."

Response: Done as suggested.

- Line 489: "... pre-tRNAs simultaneously ..."

Response: Done as suggested.

- Line 537: "The template that contains a T7 promoter sequence upstream of the pre-tRNA or RNase P RNA coding region was obtained by overlap PCR using the primers listed in Table S4."

Response: Done as suggested.

- Line 541: "To remove the hammerhead ribozyme and increase RNase P RNA yield, the DNase I-digested transcription sample was incubated overnight at 4°C."

Response: Done as suggested.

- Line 558: "Assays with a final volume of 10 μ L were set up. The reaction mixtures that included 1 μ M pre-tRNA and protein serially diluted from 1 μ M to 10 μ M were incubated ..."

Response: Done as suggested.

- Line 563: "... 4°C) in 1xTAE buffer."

Response: Done as suggested.

- Line 580: delete 'including'

Response: Done as suggested.

- Line 584: "... were 0.25 μ M, 5 μ M and 5 mM, ..."

Response: Done as suggested.

- Line 585: "The reactions were terminated after 15 min by heating to 95°C for 5 min."

Response: Done as suggested.

- Line 741: replace C) with E)

Response: Done as suggested.

Reviewer #2 (Remarks to the Author):

The authors have answered most of my questions and modified their text accordingly. I feel that the authors are honestly and carefully explaining and interpreting the new data (e.g. EMSA, nano-DSF) in the response letter. However, these explanations and interpretations are not reflected and are inadequate in the revised manuscript. Those should be clearly addressed in the manuscript.

Response: We sincerely thank the reviewer for all the nice comments and criticisms. Based on the reviewer's comments and suggestions, we have carefully revised our manuscript.

The following two points are particularly important.

1. Interpretation or explanation of discrepancies between EMSA data and cleavage assay data for the mutants

Response: We sincerely thank the reviewer for raising this interesting question. As we demonstrated in Fig. 1D and the panel A of the figure below, PbHARP is very positive in charge. Out of the total 203 amino acids of PbHARP, 37 are Arg or Lys (please see Fig. S4A and the panel B of the figure below). We speculated that some of these Arg or Lys residues can replace R116, R123, R138 or R142 in pre-tRNA binding, but the resulting complex is incompatible with the catalysis. We are very sorry that we do not have the pre-tRNA-complexed structure of the mutated PbHARP protein to directly confirm this hypothesis, but we believe that our in vitro cleavage assay results (please see Fig. 2F and the panel C of the figure below) can support the importance of the pre-tRNA recognition residues in the assembly of the catalytic PbHARP/pre-tRNA complex. We have reflected all these opinions in the revised manuscript.

2. Monomerization by the L103E/V107E mutation affects protein stability? or the L103E/V107E mutation itself affect protein stability?

Response: *We sincerely thank the reviewer for the thoughtful comments. To investigate the impact of the L103E/V107E mutation, we generated one structural model for the L103E/V107E mutant using the AlphaFold2 program. Comparison of the model and the crystal structure of PbHARP (please see Fig. S9C and the figure below) suggested that the L103E/V107E mutation will not affect the folding of PbHARP. We believed that the failure of dimerization may play important role in the lower stability of the L103E/V107E mutant. We have reflected this opinion in the revised manuscript.*

Minor point

i) The domain architectures of AtPRORP1 in Fig1 A lacks the Nuclease domain.

Response: *We sincerely thank the reviewer for pointing out our mistake. The Fig. 1A has been updated in the revised manuscript.*

ii) In the figure legend of Fig 4, there are two Cs and no E.

Response: *We sincerely thank the reviewer for pointing out our mistake, which has been fixed in the revised manuscript.*

We sincerely thank the reviewers for rereading our manuscript with great care, we would also like to thank the reviewers for their very helpful comments, encouragements and criticisms as well. On the basis of their comments and suggestions, we have carefully revised our manuscript and addressed all the remaining issues raised by the reviewers. We believe that the quality of our manuscript has been significantly improved. The following are our point-to-point responses to the reviewers' comments.

Reviewer #1 (Remarks to the Author):

The authors have addressed all experimental issues satisfactorily, which largely improved the quality of the manuscript. After careful rereading, several minor issues remain where (A) statements should be set right or rewritten for better understandability, in addition to (B) several 'language polishing' suggestions. I also proposed changes/reference additions at two locations in the Discussion to better appreciate the two recently reported cryo-EM structures of HARP dodecamers. Overall, there is little comparative discussion of the findings of the two HARP cryo-EM structures, which is still a weakness of the manuscript.

Response: We sincerely thank the reviewer for all the nice comments and criticisms. Based on the reviewer's comments, we have included one additional figure (please see Fig. S16 and the figure below) to compare our and the reported cryo-EM structure of HARP in the revised manuscript.

A. Rephrasing of statements:

- Line 69, rewrite: "Bacteria and Archaea do not express eukaryotic-like PRORP proteins, but ..."

Response: Done as suggested.

- Line 73: "... are also present in some other bacteria and are even more widespread in Archaea^{30,31}."

Response: Done as suggested.

- Line 84: add ref. 33 after 'euryarchaeotes'

Response: Thanks for the suggestion. Ref. 33 has been cited after

'euryarchaeotes' in the revised manuscript.

- Line 85: delete 'some'

Response: Done as suggested.

- Lines 93-95: delete sentence "However, likely due to their important role in tRNA maturation, no HARP gene knockout strains has been reported for any Aquificaceae." (knockouts have not been tested in members of the Aquificaceae because these bacteria are not genetically manipulable)

Response: We thank the reviewer for the thoughtful comments. The statement has been removed in the revised manuscript.

- line 102: "... into oligomers. Structural analysis, mutagenesis, in vitro cleavage and crosslinking assay all suggested a HARP pre-tRNA binding and cleavage mode involving the cooperation of two dimers."

Response: Done as suggested.

- Line 114, rewrite to: "... similarities with Aq_880, and the sequence identity is about 30% among the five proteins (Fig. S1B)." (30% identity is not necessarily very low among orthologous proteins)

Response: We thank the reviewer for the suggestion. This sentence has been rewritten in the revised manuscript.

- Line 181, rephrase: "The asymmetric dimerization orients the side chains of four HB domain Arg residues, including R116 and R123 of monomer B and R138 and R142 of monomer A, toward the pre-tRNA^{His} T ψ C-loop where they form extensive H-bond interactions with the phosphate backbones of G52-G56 (Fig. 2B)."

Response: Done as suggested.

- Line 204, replace sentence beginning with 'Although': "Two 5'-leader cleavage products were observed, one being identical in size to that generated by Aq_880 and E. coli RNase P (Fig. S3C, outer right lane). This shows that PbHARP cleaves pre-tRNA at the canonical cleavage site in addition to some apparent miscleavage at the neighboring upstream site."

Response: Done as suggested.

- Line 219, rewrite: "... are critical for assembly of the catalytically active PbHARP/pre-tRNA complex."

Response: Done as suggested.

- Line 226, rephrase to: "The functional importance of these catalytic residues could be confirmed by ..." (their catalytic importance was already shown in ref. 30)

Response: Thanks for the thoughtful suggestion. This sentence has been rephrased in the revised manuscript.

- Line 251: “Although not observed in the structure, this arrangement may allow PbHARP to coordinate a second divalent cation, thereby utilizing the common two metal ion-assisted mechanism of catalysis^{35,37,38}.”

Response: Done as suggested.

- Line 254: “Divalent metal ion coordination, ...”

Response: Done as suggested.

- Line 282: Here the authors state “However, the C-termini of $\alpha 7$ helices gathered together in the complex structure, ...”. This is not immediately evident from Fig. 2. I propose to refer here also to Fig. S5 that illustrates this well.

Response: We sincerely thank the reviewer for the great suggestion. In addition to Fig. 2, Fig. S5A and S5C have also been cited in the revised manuscript.

- Line 454, rewrite: “The total size of HARP is only about 1/3 of AtPRORP1 and HsMRPP3, the latter even requiring two additional protein cofactors (HsMRPP1 and HsMRPP2) for in vivo activity. However, HARP performs the same reaction as AtPRORP1 and the HsMRPP complex. A similar phenomenon has also been observed ...”

Response: Done as suggested.

- Line 465: “Our and the related recent studies 52,53 of HARP support the idea that oligomerization and division of function between mono- or dimers may be a more common way (than previously thought) to provide small enzymes with substrate binding and catalytic abilities.”

Response: Thanks for the suggestion. The recently reported HARP structures were mentioned here. The references were numbered 44 and 45 in the revised manuscript.

- Line 473: the ref. numbers must be 28,29,51

Response: Thanks for the correction. The references have been updated in the revised manuscript.

- Line 487: We believe that a tetramer is the minimal catalytic unit of HARP in vivo, whereas dodecamerization may allow HARP to bind and cut multiple pre-tRNAs simultaneously, as proposed recently 52,53.

(add here also ref. 53!)

Response: Thanks for the suggestion. Reference 52 and 53 have been renumbered as 44 and 45, and both references were cited at the end of the sentence in the revised manuscript.

- Line 489: "Our study reveals the molecular basis for substrate binding and cleavage by HARP proteins."

Response: Done as suggested.

- Lines 495, 501,525, 538,542, 546, 579, Table S2: the genes for RNA-based bacterial RNase P are written as *rnpA* and *rnpB* (in italics); the RNA itself is named 'RNase P RNA' or 'P RNA'; the RNase P protein should be written as 'RnpA'. Please change correspondingly.

Response: We sincerely thank the reviewer for pointing out our mistakes. Based on the reviewer's suggestions, we have carefully examined and corrected all the mistakes in the revised manuscript.

- Line 501: "... transcription of E. coli RNase P RNA, the *rnpB* gene with T7 promoter and hammerhead ribozyme (Table S2) was inserted into a pUC18 vector at the HindIII and EcoRI restriction sites."

Response: Done as suggested.

- Lines 585-587: the addition and composition of gel loading buffer is missing in the method description.

Response: Thanks for the helpful suggestion. The addition and the detailed composition of gel loading buffer have been included in the revised manuscript.

Legend to Fig. S7, line 70, rewrite: "... of divalent metal ions (Mg^{2+}) and the catalytic Asp residues of PbHARP."

Response: Done as suggested.

B. Linguistic/grammatical corrections:

- Line 99: "... showed that HARP is able to undergo large conformational changes that facilitate pre-tRNA binding and catalytic site formation."

Response: Done as suggested.

- Line 120: "... it exists as dimer; however, it assembles into larger oligomers in buffer containing 100 mM salt."

Response: Done as suggested.

- Line 149: "... forming a four-helix bundle ..."

Response: Done as suggested.

- Line 153: " $\beta 4$ plays an important ..."

Response: Done as suggested.

- Line 207: "... are functionally ..."

Response: Done as suggested.

- Line 231: "... about 60% of the substrate were cleaved; only trace amounts of intact substrate remained ..."

Response: Done as suggested.

- Line 238: "... In contrast, the side chain of D151 points away from D155, the averaged distance between their carboxylate groups exceeding 7 Å."

Response: Done as suggested.

- Line 244: "... As demonstrated for many nucleases, ..."

Response: Done as suggested.

- Line 264: "... $\alpha 8$ is split into ..."

Response: Done as suggested.

- Line 285: "In the complex structure, the A118 residues center around the HB domain central axis, the distance between their side chains is only 3.6 Å (Fig. 4C)."

Response: Thanks for the suggestion. Fig. 4C has been cited at the end of this sentence in the revised manuscript.

- Line 296: "... we also constructed the A118E/A121E mutant ..."

Response: Done as suggested.

- Line 300: "... is weak; no pre-tRNA binding at all was observed ..."

Response: Done as suggested.

- Line 302: "... but detectable pre-tRNA^{His} cleavage activity was neither observed for the L103E/V107E nor for the A118E/A121E mutant under identical reaction conditions (Fig. 4E), probably due to their low stability and/or difficulty to form a functional dimer."

Response: Done as suggested.

- Line 308: "In the structure of the complex, PbHARP mainly recognizes the shape and backbone of the T ψ C-loop (Fig. 2B-E), a structural element conserved in all tRNAs."

Response: Done as suggested.

- Line 313: "... 5'-end maturation, the phosphodiester between nucleotides -1 and +1 is usually cleaved."

Response: Done as suggested.

- Line 321: "... is much weaker than those of His-7bp and His-6bp. In fact, the cleavage efficiency of His-7bp was comparable to those of ..."

Response: Done as suggested.

- Line 329: "Both TrHARP and PbHARP preferably cleaved the His-7bp variant (Fig. 5B-C), but their oligomerization states are different. TrHARP eluted from the size exclusion column at ~11.4 mL in all buffers containing 100, 200 or 500 mM salt (Fig. 6A). A previous study suggested the possibility that Aq_880 may form either six trimers or three hexamers³⁰ (inferred from size exclusion chromatography profiles), but both PbHARP and TcHARP formed dimers in the crystal structures (Fig. 1C-F)."

Response: Done as suggested.

- Line 335: "... was subjected ..."

Response: Done as suggested.

- Line 340: "... shape and size of the ring matched well with an assembly of six PbHARP dimers that arrange in a left-handed orientation (Fig. 6D)."

Response: Done as suggested.

- Line 347: "... are ~50° between Dimer_1 and Dimer_2, Dimer_2 and Dimer_3, and ..."

Response: Done as suggested.

- Line 360: "... with TrHARP suggested that PbHARP has the potential to form larger oligomers under proper conditions. The incorporation of additional PbHARP dimers can be easily modeled by rotating ..."

Response: Done as suggested.

- Line 372: "G-1 pairs with C72 in native pre-tRNA^{His}, and such pairing may interfere with the conformational change and metal ion coordination at the scissile phosphodiester between G-1 and G1. The acceptor stem of His-9bp is 2 bp longer ..."

Response: Done as suggested.

- Line 386: "... had no strong impact on ..."

Response: Done as suggested.

- Line 389: "Pre-tRNA^{His} is recognized by the PbHARP AB dimer in the complex, but neither R108 nor R146 of the AB dimer are involved in the interaction. These structural observations combined with the inactivity of these mutants further suggested that PbHARP binds and cleaves pre-tRNA in a cooperative mode. To better demonstrate the cooperation between PbHARP dimers, we performed ..."

Response: Done as suggested.

- Line 394: "In addition to the monomer, ..."

Response: Done as suggested.

- Line 397: "...of two lysine residues. Although not clearly detectable in the absence of His-7bp, ..."

Response: Done as suggested.

- Line 407: "However, unlike PRORP enzymes constituting the PRORP subfamily, HARPs belong to the PIN_5 cluster subgroup."

Response: Done as suggested.

- Line 415: "The activities of HARP ..."

Response: Thanks for the correction. The word "activity" has been changed into it's plural form.

- Line 416: "The apo-form HsMRPP3 structure is in an inactive form, ..."

Response: Done as suggested.

- Line 419: "... and TcHARP are also in an inactive form."

Response: Done as suggested.

- Line 421: "... suggested that HARP undergo self-activation via a large conformational change to the catalytically active form (Fig. 3B-D)."

Response: Done as suggested.

- Line 426: " However, the third catalytic Asp residue (D151) of PbHARP is positioned in mirror image to D399 of AtPRORP1 (Fig. S12E). This mirror-image position of D151/D399 and the self-activation ability distinguish HARP from eukaryotic PRORPs."

Response: Done as suggested.

- Line 436: "... HARP and AtPRORP1 PPR both recognize ..."

Response: Done as suggested.

- Line 439: "... helices of the HARP HB and the AtPRORP1 PPR domain ..."

Response: Done as suggested.

- Line 445: "... central domain at the bottom. Although it has not been experimentally verified, it is likely that the V-shaped conformation ..."

Response: Done as suggested.

- Line 449: "Neither a single monomer nor dimer of HARP ..."

Response: Done as suggested.

- line 451: "... results suggest that ..."

Response: Done as suggested.

- Line 452: "Interestingly, although not identical to AtPRORP1, the two HARP dimers also seem to adopt a V-shaped conformation (Fig. 7B)."

Response: Done as suggested.

- Line 459: "... functions as a homodimer."

Response: Done as suggested.

- Line 474: "... archaea and bacteria that also encode ..."

Response: Done as suggested.

- Line 480: "... assembled into dodecameric structures. The PbHARP and TcHARP structures we report here are the only available crystal structures of HARP."

Response: Done as suggested.

- Line 484: "Instead of being dodecamers, PbHARP and TcHARP exist ..."

Response: Done as suggested.

- Line 489: "... pre-tRNAs simultaneously ..."

Response: Done as suggested.

- Line 537: "The template that contains a T7 promoter sequence upstream of the pre-tRNA or RNase P RNA coding region was obtained by overlap PCR using the primers listed in Table S4."

Response: Done as suggested.

- Line 541: "To remove the hammerhead ribozyme and increase RNase P RNA yield, the DNase I-digested transcription sample was incubated overnight at 4°C."

Response: Done as suggested.

- Line 558: "Assays with a final volume of 10 μ L were set up. The reaction mixtures that included 1 μ M pre-tRNA and protein serially diluted from 1 μ M to 10 μ M were incubated ..."

Response: Done as suggested.

- Line 563: "... 4°C) in 1xTAE buffer."

Response: Done as suggested.

- Line 580: delete 'including'

Response: Done as suggested.

- Line 584: "... were 0.25 μ M, 5 μ M and 5 mM, ..."

Response: Done as suggested.

- Line 585: "The reactions were terminated after 15 min by heating to 95°C for 5 min."

Response: Done as suggested.

- Line 741: replace C) with E)

Response: Done as suggested.

Reviewer #2 (Remarks to the Author):

The authors have answered most of my questions and modified their text accordingly. I feel that the authors are honestly and carefully explaining and interpreting the new data (e.g. EMSA, nano-DSF) in the response letter. However, these explanations and interpretations are not reflected and are inadequate in the revised manuscript. Those should be clearly addressed in the manuscript.

Response: We sincerely thank the reviewer for all the nice comments and criticisms. Based on the reviewer's comments and suggestions, we have carefully revised our manuscript.

The following two points are particularly important.

1. Interpretation or explanation of discrepancies between EMSA data and cleavage assay data for the mutants

Response: We sincerely thank the reviewer for raising this question. To resolve the discrepancies between EMSA data and cleavage assay data, we did careful sequence and structural analysis of PbHARP. We found that PbHRAP is longer at its N-terminus, compared to many other homology proteins. The extra PbHRAP terminus is highly positive in charge; out of the total 13 residues, 7 are Arg or Lys (Fig. S1B). We speculated that the N-terminal residues of PbHARP may be able to substitute the pre-tRNA recognition residues (R116, R123, R138 and R142) in interacting with pre-tRNA, leading to the weird binding assay results. To test this hypothesis, we constructed one truncated protein, PbHARP_ΔN (aa 14-203). Compared to the full-length protein, the pre-tRNA binding ability of PbHARP_ΔN is weaker. Started from PbHARP_ΔN, we then constructed several PbHARP mutants with pre-tRNA recognizing residue mutation. Although it has no strong impact on the overall pre-tRNA binding ability, Ala substitution of either R116 (for the PbHARP_ΔN_R116A mutant) or R123 (for the PbHARP_ΔN_R123A mutant) altered the pre-tRNA binding mode of PbHARP, indicated by the smeared bands on the gel. The pre-tRNA binding ability of PbHARP_ΔN_R138A is weaker than that of PbHARP_ΔN. No obvious difference was observed for the PbHARP_ΔN_R142A mutant, but PbHARP_ΔN_R138A/R142A mutant showed weaker pre-tRNA binding ability, compared to that of the PbHARP_ΔN_R138A mutant. All together, these mutation and binding assay results suggested that R116, R123, R138 and R142 are important for pre-tRNA binding by PbHARP. The new results have been included in the revised manuscript (Please see the figure below and Fig. S6B in the revised manuscript).

2. Monomerization by the L103E/V107E mutation affects protein stability? or the L103E/V107E mutation itself affect protein stability?

Response: *We sincerely thank the reviewer for the thoughtful comments. To better understand the impact of the L103E/V107E mutation, we performed CD analysis for the WT and the L103E/V107E mutant of PbHARP. In consistent with the crystal structure, the CD results showed that the WT PbHARP protein possesses ~50% α -helices and 10% β -strands (Parallel + Antiparalel) at 20°C; the calculated T_m value is ~70.0°C. Different from the WT protein, the L103E/V107E mutant possesses ~30% α -helices and 20% β -strands (Parallel + Antiparalel) at 20°C; the T_m value of the mutant is about 53.0°C. These results suggested that the L103E/V107E mutation affected the proper folding of PbHARP. Instead of monomerization, we believed that misfolding may play more important role in the lower stability of the L103E/V107E mutant. These new results have been included in the revised manuscript and showed in the figure below.*

Minor point

i) The domain architectures of AtPRORP1 in Fig1 A lacks the Nuclease domain.

Response: *We sincerely thank the reviewer for pointing out our mistake. The Fig. 1A has been updated in the revised manuscript.*

ii) In the figure legend of Fig 4, there are two Cs and no E.

Response: *We sincerely thank the reviewer for pointing out our mistake, which has been fixed in the revised manuscript.*

REVIEWERS' COMMENTS

Reviewer #2 (Remarks to the Author):

The authors have struggled to resolve the discrepancies between EMSA and cleavage assay data. They provided new data showing that the mutants with N-terminal deletion altered the pre-tRNA binding mode. These results are interesting. They have addressed my concerns and described their explanations and interpretations in the revised manuscript.

After careful reading, several minor issues remain.

In181-183: G52-G56, G52-C55 might be mistaken for base pairs, so please add 'region' after them. Ex. G52-G56 region

Fig.S3C: The miscleavage 5'-leader fragment of pbHARP should also be indicated by an arrow.

Fig. S5C: It is hard to tell which color is which monomer. Please add color legends.

Fig. S6: The authors discussed the strength of affinity, but they provided no quantitative affinity constant. Please provide the affinity values in the figure.

REVIEWERS' COMMENTS

Reviewer #2 (Remarks to the Author):

The authors have struggled to resolve the discrepancies between EMSA and cleavage assay data. They provided new data showing that the mutants with N-terminal deletion altered the pre-tRNA binding mode. These results are interesting. They have addressed my concerns and described their explanations and interpretations in the revised manuscript.

Response: We sincerely thank the reviewer for rereading our manuscript with great care, we would also like to thank the reviewer for all the nice comments and encouragements.

After careful reading, several minor issues remain.

In181-183: G52-G56, G52-C55 might be mistaken for base pairs, so please add 'region' after them. Ex. G52-G56 region

Response: Done as suggested.

Fig.S3C: The miscleavage 5'-leader fragment of pbHARP should also be indicated by an arrow.

Response: Done as suggested.

Fig. S5C: It is hard to tell which color is which monomer. Please add color legends.

Response: Done as suggested.

Fig. S6: The authors discussed the strength of affinity, but they provided no quantitative affinity constant. Please provide the affinity values in the figure.

Response: We sincerely thank the reviewer for the helpful suggestion. The K_d values have been included in the figure.